

# Optimal Inverse Estimation of Ecosystem Parameters from Observations of Carbon and Energy Fluxes

Debsunder Dutta[1], David S. Schimel[1], Ying Sun[3], Christiaan van der Tol[4], and Christian Frankenberg[1,2]

[1]Jet Propulsion Laboratory, California Institute of Technology, Pasadena, CA, United States.
[2]Division of Geological and Planetary Sciences, California Institute of Technology, Pasadena, CA, , United States.
[3]School of Integrative Plant Science, Soil and Crop Sciences Section, Cornell University, Ithaca, NY, United States.
[4]Faculty of Geo-Information Science and Earth Observation (ITC), University of Twente, Enschede, The Netherlands.

**Correspondence:** Debsunder Dutta (Debsunder.Dutta@jpl.nasa.gov)

**Abstract.** Canopy structural and leaf photosynthesis parameterizations such as maximum carboxylation capacity ($V_{cmax}$), slope of the Ball-Berry stomatal conductance model ($BB_{slope}$) and leaf area index (LAI) are crucial for modeling the plant physiological processes and canopy radiative transfer. These parameters are large sources of uncertainty in predictions of carbon and water fluxes. In this study, we develop an optimal inversion framework to use the Soil Canopy Observation Photo-

chemistry and Energy fluxes (SCOPE) model for estimating $V_{cmax}$, $BB_{slope}$ and LAI by constraining observations of coupled carbon and energy fluxes from eddy covariance towers. We adapted SCOPE to follow the biochemical implementation of the Community Land Model and applied a moving window Bayesian non-linear inversion framework using SCOPE to invert the ecosystem parameters $V_{cmax}$, $BB_{slope}$ and LAI that best match flux-tower observations of Gross Primary Productivity (GPP) and Latent Energy (LE) fluxes. We applied this inversion framework to plant species having both the $C_3$ and $C_4$ photosynthetic

pathways across three different ecosystems. Our results demonstrate the applicability of the approach in terms of capturing the seasonal variability and posterior error reduction (40-90%) of key ecosystem parameters. The optimized parameters capture the diurnal and seasonal variability in the GPP and LE fluxes well when compared to flux tower observations ($0.95 > R^2 > 0.79$). This study thus demonstrates the feasibility of parameter inversions using SCOPE, which can be easily adapted to incorporate additional data sources such as spectrally resolved reflectance and solar induced chlorophyll fluorescence.

# 1  Introduction

Terrestrial ecosystems play a very important role in regulating the carbon exchange over land surfaces(Schimel, 1995; Falkowski et al., 2000). Although they are known to be important sinks in buffering the increasing anthropogenic $CO_2$ emissions (Friedling-stein et al., 2006; Sitch et al., 2015), there is a large variability and heterogeneity in the carbon exchange mechanisms which are tightly correlated with inter-annual climatic variations (Cox et al., 2013; Liu et al., 2017). Moreover, terrestrial ecosys-

tems also control the exchange of energy, water and momentum between the atmosphere and the land-surface, thus regulating climate-ecosystem (carbon) feedbacks leading to amplification or dampening of regional and global climate change (Heimann and Reichstein, 2008). Measurements and modeling of carbon and water vapor fluxes over terrestrial ecosystems are therefore extremely important to better understand these issues and account for the regional and global carbon and water budgets (Bal-





docchi et al., 1996, 2001; Sitch et al., 2003, 2008). Terrestrial ecosystem models have been used to study the carbon and water fluxes (McGuire et al., 2001; Sitch et al., 2003; Cramer et al., 2001; Kucharik et al., 2000), however there are large uncertainties in fluxes associated with poorly quantified model parameters (Knorr and Heimann, 2001; Zaehle et al., 2005; Rogers et al., 2017). Some of these parameters have temporal and spatial variability and are hard to measure directly over large scales

(Simioni et al., 2004; Wilson et al., 2000; Dutta et al., 2017). For the majority of model implementations these parameters and their temperature dependence are represented as a single constant value according to plant functional types with little or no seasonal variability. In this study, we present an inversion approach which can be implemented with ecosystem models involving canopy physiological processes to better estimate the seasonal variability in photosynthesis and canopy structural parameters, which in turn can reduce the uncertainty in estimation of carbon and water fluxes over ecosystems.

The micrometeorological data from flux towers is extremely useful in understanding the biogeochemistry and thermodynamics of ecosystems (Baldocchi et al., 2000; McGuire et al., 2002). A number of approaches have been developed to model and estimate photosynthesis, respiration, energy balance, stomatal behavior, radiation transfer and turbulent gas exchange across the plant canopy on the basis of data from flux tower experiments (van der Tol et al., 2009; Running and Coughlan, 1988; Oleson et al., 2010). Detailed canopy models are often resolved into multiple layers thus providing a better treatment of radiation

regime and energy balance across the canopy (van der Tol et al., 2009; Wang and Leuning, 1998; Dai et al., 2004). At the heart of these models lies the leaf level biochemical model of photosynthesis and carbon fixation (Farquhar et al., 1980) together with a stomatal conductance (most often the widely used Ball-Berry) model (Collatz et al., 1991b). The fluxes of carbon and water are tightly coupled through stomatal regulation and photosynthesis (Baldocchi, 1994; Collatz et al., 1992). These models require some environmental drivers such as incoming photosynthetically active radiation (PAR), air temperature, relative

humidity, wind speed, and ambient $CO_2$ concentration, along with a few leaf and canopy parameters to simulate the fluxes of carbon in terms of gross primary production (GPP), flux of water or latent energy (LE), sensible heat (H), net radiation and others.

     One of the most important ecosystem descriptors is the maximum rate of carboxylation ($V_{cmax}$), which is directly related to the concentration of the enzyme Rubisco. $V_{cmax}$ is a key parameter in the equation representing the Michaelis-Menten kinetics

for an enzyme-catalyzed reaction of the substrates $CO_2$ or $O_2$ with ribulose-1,5-bisphosphate, representing the enzyme-limited photosynthesis rate (Farquhar et al., 1980). Other rate-limiting photosynthesis parameters such as maximum electron transport rate ($J_{max}$) and dark respiration ($R_d$) are generally parameterized with respect to $V_{cmax}$. The Ball-Berry equation calculates the stomatal conductance ($g_s$) for water vapor as a function of net assimilation, relative humidity, leaf surface $CO_2$ concentration, minimum conductance and a proportionality constant called the Ball-Berry slope ($BB_{slope}$) (Wullschleger, 1993; Beerling and

Quick, 1995; Tanaka et al., 2002). The $BB_{slope}$ is also one of the most important ecosystem parameters as it plays a crucial role in regulating the stomatal conductance and water use efficiency, and thus the surface energy fluxes in terms of partitioning the turbulent energy into LE and H fluxes. Thus, it is a crucial parameter regulating the tradeoff between carbon gain and water loss, e.g. during drought conditions (Monteith and Unsworth, 2007). The Leaf Area Index (LAI) is a canopy structural and key ecosystem parameter which determines interception of radiation as well as photosynthesis and energy exchange across the

canopy (Chen et al., 1997). The parameters $V_{cmax}$ and $BB_{slope}$ can be determined experimentally from leaf level gas exchange



measurements and generated A-C$_i$ curves (Wullschleger, 1993; Tanaka et al., 2002; Xu and Baldocchi, 2003). LAI can be estimated from non-destructive optical methods (Myneni et al., 1997; Dutta et al., 2017; Chen et al., 1997), as well as inversion approaches on spectrally resolved reflectance data from satellite and airborne platforms (Houborg et al., 2007; Jacquemoud et al., 1995). However, these measurements are much more complex and labor intensive, being measured much less frequently

than flux tower observations.

Inversion of detailed process-based models using observations of carbon and energy fluxes could thus yield these key ecosystem parameters. Process based models such as the Soil Canopy Observation, Photochemistry and Energy fluxes (SCOPE) (van der Tol et al., 2009) can simulate the radiative transfer and the fluxes of carbon and energy vertically resolved within the canopy. Our hypothesis is that the inversion of detailed vertically resolved canopy model such as SCOPE with multiple

layers consisting of sunlit and shaded fractions together with fully spectrally resolved radiation regime and energy balance computations (van der Tol et al., 2009) is able to retrieve the ecosystem parameters accurately using observations of carbon and energy fluxes, and in the future remote sensing data, as SCOPE can model the spectrally resolved short-wave reflectance, thermal emission and solar induced chlorophyll fluorescence. It can be noted that LAI is a parameter in SCOPE, not a state variable as in dynamic vegetation models.

A few studies have used inversion approaches to extract ecosystem parameters from flux measurements (Reichstein et al., 2003) but not yet to constrain all three key parameters (V$_{cmax}$, BB$_{slope}$ and LAI) simultaneous using the fluxes of water and carbon (Schulze et al., 1994). A previous study by Wolf et. al (Wolf et al., 2006) used deterministic linear least-squares inversion method to estimate the key ecosystem parameters (V$_{cmax}$, BB$_{slope}$, LAI and respiration rate) using the net ecosystem exchange (NEE) and sensible and latent heat fluxes. The approach assumed a simple model of radiation driven photosynthesis,

respiration and energy balance using a two component (sunlit and shaded) canopy. The optimization used total energy (H+LE) to fit LAI values, the NEE to fit V$_{cmax}$ and respiration rate and energy difference (H-LE) to fit BB$_{slope}$. In comparison to deterministic approaches the stochastic Monte-Carlo approach (Knorr and Kattge, 2005; Xu et al., 2006; Ricciuto et al., 2008) constrains a number of parameters using eddy covariance observations but assuming them to be time invariant, including the photosynthetic parameters. Moreover, since the stochastic methods sample the probability distribution in parameter space, they

are better suited to non-linear models but the associated computational costs can be prohibitive.

In this study, we develop an inversion framework for estimating key ecosystem parameters using the SCOPE model representing detailed plant physiological processes including Sun Induced chlorophyll Fluorescence (SIF). SIF is chlorophyll re-emission during photosynthesis and acts as a direct probe into photosynthesis measurable from space and is strongly correlated with flux based GPP estimates at canopy to ecosystem scales (Frankenberg et al., 2011; Flexas et al., 2002). Thus, the

SCOPE based inversion approach has the flexibility and advantage of incorporating tower-based observations of fluxes including SIF as well as spectrally resolved reflectance for optimal estimation of a wide range of ecosystem parameters. However, in this paper, we first focus on the conceptual framework of parameter inversion using SCOPE, with specific objectives as follows:

1. Implementation of photosynthesis model and its temperature dependencies consistent with a well-accepted major Earth
system model (Community Land Model CLM 4.5) in SCOPE.





2. Development of a Bayesian non-linear inversion framework using SCOPE to estimate ecosystem parameters using eddy covariance flux observations.

3. Demonstrating the retrieval and posterior error reduction of key ecosystem parameters using observations of carbon and water fluxes across different ecosystems.

The rest of the paper is organized as follows. Section 2 provides a brief overview of the SCOPE model and the new implementation of photosynthesis and its temperature dependencies. Section 2.3 provides a comparison of the old and new photosynthesis implementations in SCOPE. Sections 3, 4 and 5 describes the formulation of the inverse problem followed by linearization of the forward model and mechanisms of the retrieval algorithm. Section 6 describes the results of the inversion framework across three different ecosystems and finally Section 7 provides a discussion summary and conclusions.

## 2 SCOPE Model

The Soil, Canopy, Observation, Photochemistry and Energy fluxes (SCOPE) (van der Tol et al., 2009) is an integrated 1-D vertical radiative transfer and energy balance model. The model utilizes the spectrally resolved visible to thermal (0.4 to 50 $\mu$m) infrared irradiation at the canopy top to derive the fluxes of water, energy, carbon dioxide and vertical profiles of temperature as a function of canopy structure and weather variables. The four most important SCOPE modules represent (i) radiative transfer of incident solar radiation and generated fluorescence within the leaf (Fluspect), (ii) radiative transfer of incident direct and indirect solar radiation (0.4 - 50 $\mu$m), (iii) radiative transfer of internally generated thermal radiation by vegetation and soil (Verhoef et al., 2007), (iv) an energy balance module (EBAL) and (v) radiative transfer module for computing the top of canopy radiance spectrum of fluorescence from leaf level chlorophyll fluorescence.

The leaf radiative transfer model computes the leaf reflectance, transmittance, bi-directional fluorescence emission, and the absorbed PAR per pigment. The solar radiative transfer model computes the top of canopy (TOC) outgoing radiance spectrum as well as net radiance and absorbed PAR per surface elements, the thermal radiative transfer module computes the TOC outgoing thermal radiation and net radiation per surface element for heterogeneous leaf and surface temperatures generated internally by soil and vegetation, the EBAL computes the latent, sensible and soil heat per element as well as photosynthesis, fluorescence and skin temperatures, finally the fluorescence radiative transfer module computes the outgoing top of canopy radiance spectrum of fluorescence.

One important aspect is that SCOPE relaxes the assumption of constant temperatures for the sunlit and shaded fractions of the leaves across the different canopy layers. This is true when we consider different orientations, and their vertical positions in the canopy. Therefore, an iterative solution scheme is implemented in SCOPE as stomatal conductance affects leaf temperature, which in turn affects photosynthesis (and thus again stomatal conductance). Thus, the fully integrated thermal radiative transfer and EBAL modules allow feedback between leaf temperatures, photosynthesis, chlorophyll fluorescence, and radiative fluxes.

Within the heart of the EBAL module (and also essentially SCOPE) is the biochemical module for the computation of photosynthesis and chlorophyll fluorescence at the leaf level. Photosynthesis and stomatal regulation is one of the most important




physiological processes which controls each of the outgoing fluxes and radiances. Therefore, its computation and temperature dependence are crucial for accurate estimation of the canopy net fluxes.

## 2.1 The SCOPE Biochemical Module

The SCOPE biochemical module is a submodule of the EBAL routine which provides an iterative solution of the photosynthe-
sis, energy balance, net radiation and heterogeneous skin temperatures for a particular net external forcing. The main functions
of the biochemical module include leaf temperature dependent computation of photosynthesis and fluorescence. Some of the
photosynthesis parameterizations in the current version of the SCOPE model (V1.7) are outdated and more in line with pre-
vious versions of the community land surface model (CLM4) or based on a mix of other model implementations. The main
inconsistencies with the CLM4.5 parameterizations are as follows:

1. Similar, generic temperature response functions are implemented for both $C_3$ and $C_4$ species excepting $V_{cmax}$ and further
   it uses a $Q_{10}$ based exponential function with same functional parameters for computing the temperature response of the
   various photosynthetic parameters.

2. There is no $J_{max}$ (maximum potential electron transport rate (ETR)) or its temperature dependence in the computation
   of light limited $C_3$ photosynthesis rate.

3. The net assimilation, internal $CO_2$ concentration and stomatal conductance $(A - C_i - g_s)$ iterative solution method is not
   quite robust or was lacking in the previous versions with the V1.7 implementation being complicated and unpublished.

We attempt to improve the SCOPE biochemical module by implementing the photosynthesis and temperature dependence of
the photosynthetic parameters according to well established and widely used Community Land Model (CLM V4.5 or CLM4.5)
(Lawrence et al., 2011; Oleson et al., 2013). The CLM is a community-developed land model which focus on the modeling of
land surface processes including biogeophysics, carbon cycle, vegetation dynamics and river routing. CLM is the land surface
component of the Community Earth System Modeling framework. Specifically, the main modifications in CLM4.5 include
updates to the canopy radiation scheme and canopy scaling of leaf processes, co-limitations on photosynthesis, revisions
to photosynthetic parameters (Bonan et al., 2011), temperature acclimation of photosynthesis and improved stability of the
iterative solution in the photosynthesis and stomatal conductance model (Sun et al., 2012). CLM4.5 implements a multi-layer
canopy modeling framework with coupled photosynthesis(Farquhar et al., 1980) and Ball-Berry stomatal conductance models
similar to the SCOPE framework. We therefore make the implementation of photosynthesis and its temperature dependence in
SCOPE fully consistent with CLM4.5.

All the detailed implementation steps and equations for modeling the photosynthesis and temperature dependence primarily
as per (Bonan et al., 2011) is presented in detail in the appendix A. The major new updates made to the model (biochemical
module) are as follows:





1. Computing the electron limited photosynthesis rate $A_j$ using the potential ETR $J$, which is obtained by solving the smaller root of equation A5 comprising the light utilized in photosystem II ($I_{PSII}$) and the maximum potential ETR ($J_{max}$).

2. The light limited photosynthesis rate for $C_4$ is given by equation A3, in the earlier SCOPE version it was implemented as potential ETR x $CO_2$ per electron.

3. The temperature dependence of photosynthetic parameters (Bonan et al., 2011) now uses the activation, deactivation energies and entropy terms in the temperature response and high temperature inhibition functions (Leuning, 2002) (see appendix B for details). The temperature response of $C_3$ (Leuning, 2002; Bernacchi et al., 2001) and $C_4$ photosynthesis is represented by equations B1 - B5.

4. Finally we also incorporate a new simplified implementation of $A - C_i - g_s$ iterations (Sun et al., 2012) and include the computation of oxidative photosynthesis (Bernacchi et al., 2001) within the photosynthesis model.

## 2.2 A-Ci-$g_s$ Iterations

The final solution for photosynthesis requires an iterative solution of the coupled equations representing (i) the Farquhar, von Caemmerer and Berry (FvCB) model (Farquhar et al., 1980) for the photosynthesis rate ($A$), (ii) Fick's law of diffusion (Eqn. 1) for internal $CO_2$ ($C_i$) concentration and (iii) Ball-Berry stomatal conductance model (Ball et al., 1987) (Eqn. 2) for stomatal conductance ($g_s$) to obtain stable converging solutions.

$$C_i = C_s - 1.6\frac{A}{g_s}, \quad \text{(from Fick's Law)} \tag{1}$$

$$g_s = g_0 + \text{BB}_{\text{slope}}\frac{A r_h}{C_s}, \quad \text{(Ball-Berry model)} \tag{2}$$

In Eqn. 2, $\text{BB}_{\text{slope}}$ represents the Ball-Berry slope, $r_h$ the relative humidity and $g_0$ the Ball-Berry intercept. In the absence of an initial specification of $C_i$, we make the assumption that $g_0 = 0$ in Eqn. 2, then combining equations 1, 2, the initial estimate of $C_i$ is given as:

$$C_i = \max(f_{Ci}^{min} C_s, C_s - 1.6\frac{C_s r_h}{\text{BB}_{\text{slope}}}) \tag{3}$$

where $f_{Ci}^{min}$ is the assumed minimum fractional leaf boundary layer $CO_2$ (assumed as 0.3 for $C_3$ and 0.1 for $C_4$ species). This initial estimate of $C_i$ is used to again estimate the photosynthesis based on the FvCB model (Farquhar et al., 1980), followed by estimation of stomatal conductance using the Ball-Berry model Eqn. 2. Finally, the Newton-Raphson method is used to obtain a forward estimation of the new value of internal $CO_2$ concentration (Sun et al., 2012). The updated $C_i$ is further used in the $A - g_s - C_i$ until convergence.




In the current SCOPE implementation we replaced the photosynthesis model to be fully consistent with CLM4.5. In the following section, we demonstrate the photosynthesis results with this new implementation as well as its comparison with the previous version for different ecosystems.

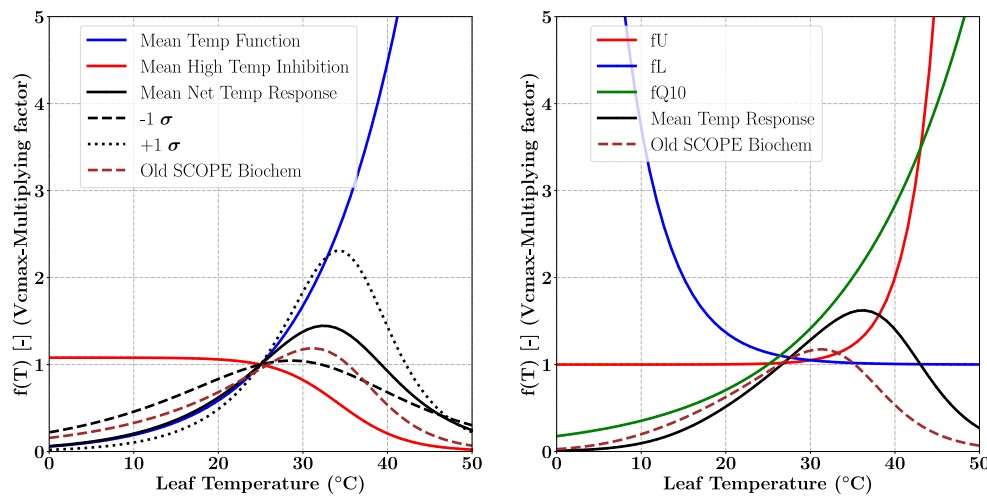

**Figure 1.** Temperature response functions of $V_{cmax}$ for $C_3$ (left) and $C_4$ (right) plants. For the $C_3$ species the $\pm 1\sigma$ variability in the net temperature response (with data from (Leuning, 2002)) is shown as broken lines in the left panel. The temperature range corresponding to maximum $V_{cmax}$ response for both the $C_3$ and $C_4$ pathways is between 30-40 $^\circ$C. The overall temperature response from the previous version (V1.7) is shown as brown dashed line.

## 2.3 Comparison of Old and New Photosynthesis Implementations in SCOPE

Figure 1 shows the temperature response functions for $V_{cmax}$ for both $C_3$ (left) and $C_4$ (right) photosynthetic pathways. The functions of mean temperature response, high temperature inhibition and the $1\sigma$ variance as per the different photosynthesis pathway dependent parameterization is shown according to (Leuning, 2002). The new temperature dependency parameterizations follow the temperature functions and high temperature inhibition for $C_3$ and the $Q_{10}$ functions for the $C_4$ pathways. We have also shown the temperature dependence of $V_{cmax}$ from the previous implementation SCOPE model (V1.70). The

differences in the net response at both lower and higher than optimal temperature can be clearly identified in the figure for both $C_3$ and $C_4$ species. It can be observed that the difference in temperature response is more for $C_4$ , clearly the maximum is in the leaf temperature range $30-40^\circ$C, however it continues into the higher temperatures as well. Moreover, it can be noted that the overall shapes of the response functions are nearly identical (with some lag) for the different parameters for the previous SCOPE implementation compared to the newer implementation as per CLM4.5 (Bonan et al., 2011).

To study the overall or net response from the SCOPE model in terms of canopy level fluxes, we tested the old and new model version for both $C_3$ and $C_4$ species at two different sites. Figure 2 shows the comparison of SCOPE model predictions for both model versions using $C_4$ plants. The results shown are for corn ($C_4$ ) crops at the Mead-1 Ameriflux site in Lincoln,





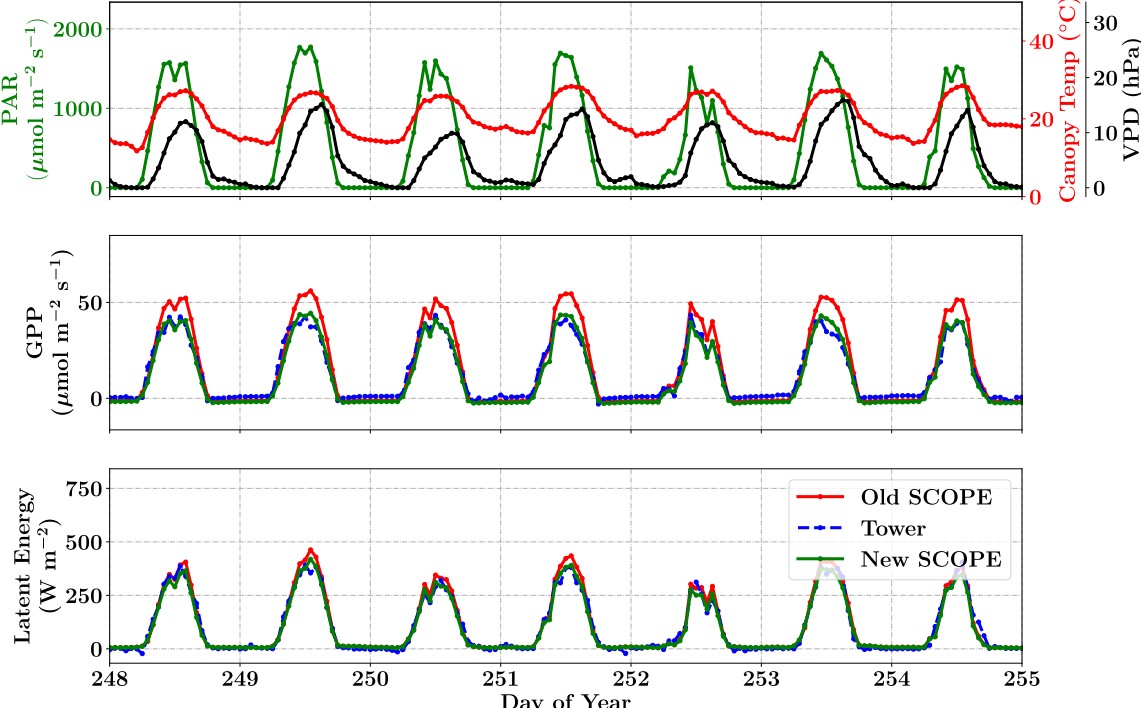

**Figure 2.** Figure showing the diurnal variability in SCOPE model simulations for $C_4$ corn species at the Nebraska, Mead-1 site for a week in the growing season for the year 2009. The top panel shows the variability of a few important meteorological forcing variables. The middle and lower panels show the variability in the diurnal cycle of GPP and latent energy respectively as a comparison between the old and new photosynthesis implementations (for identical parameterization and forcings) in SCOPE and the actual tower observed values.

Nebraska (see section 6.2.1 for some more site information). The environmental forcings and SCOPE parameterizations were kept identical for the old and new SCOPE runs. The only difference is the replacement of the old biochemical module with our new photosynthesis implementation. The top panel shows the environmental forcing for a few clear days in the growing season. There is considerable variability in temperature and VPD. The middle and lower panels represent the comparisons in diurnal GPP and LE fluxes. It is observed that with the given parameterization the newer implementation better captures the diurnal variability as compared to the actual tower observations at relatively higher average canopy temperatures.

Figure 3 shows the comparison between the old and new SCOPE as the ratio of overall canopy GPP computed using the new implementation of photosynthesis to the old implementation. The ratio is defined as ($f_{GPP} = \frac{GPP_{new}}{GPP_{old}}$). This ratio is further represented as a function of the three most important forcing variables PAR, canopy temperature and VPD. The results for the Missouri Ozark site (see section 6.3.1 for site details) with $C_3$ plant species for the year 2009 are presented in Fig. 3. For this analysis, the SCOPE model simulations are computed for the entire growing season and the $f_{GPP}$ values are binned according to the PAR-Temperature (for specific VPD ranges) and PAR-VPD (for specific temperature ranges) as 2-D histograms. Each



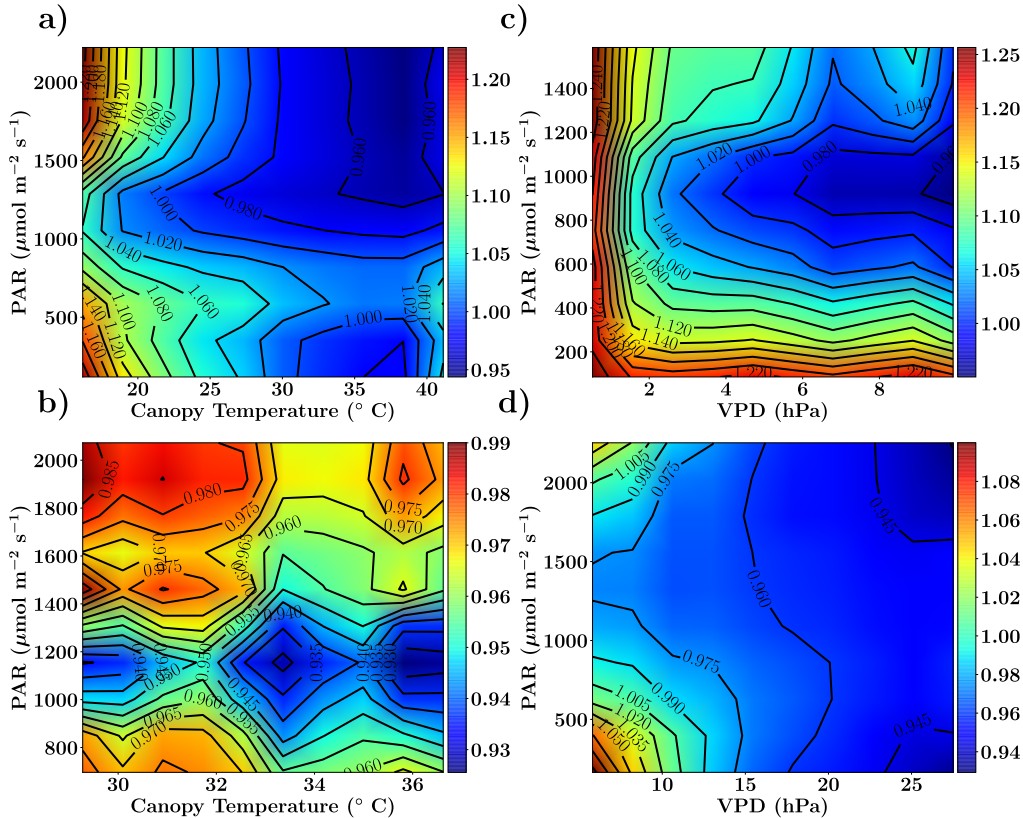

**Figure 3.** Figure showing the ratio of old and new SCOPE GPP ($f_{GPP} = \frac{GPP_{new}}{GPP_{old}}$) simulations as a function of PAR, canopy temperature and VPD for the $C_3$ species. The Missouri Ozark fluxnet site comprising of deciduous broadleaf forests for the year 2009 is used and the SCOPE simulations are driven with identical forcings and parameters for both the new and old simulations, the only difference being the implementation of Photosynthesis and its temperature dependence (see text). The left column shows $f_{GPP}$ as a function of PAR and canopy temperature with data points in the low VPD range of 10-15 hPa (panel-a) and high VPD range of 25-30 hPa (panel-b). The right column shows $f_{GPP}$ as a function of PAR and VPD with data points in the low temperature range of 10-15 °C (panel-c) and moderate temperature range of 25-30 °C (panel-d).

of these bins have multiple data points and essentially follow a distribution, of which only the mean is represented in figure 3. Finally contour plots are developed with these mean values to demonstrate the effect of paired random variables PAR-canopy temperature (shown in the left column) and PAR-VPD (shown in the left column) on $f_{GPP_{mean}}$.

We find that over the larger parts of the domain of random variables, $f_{GPP}$ is around 1 and the maximum change in overall
5    GPP is around 25%. From Figure 3, it can be observed that in the case of $C_3$ species, for the combinations of higher canopy temperature and low VPD values (panel-a), the new GPP values remain the same or are reduced by about 5%. Although from Fig. 1 we find an increase in the $V_{cmax}$ response at $> 25°C$ temperature, which may indicate photosynthesis being limited by light instead of the enzyme rubisco. For the combination of low canopy temperature and lower VPD values (panel-c),





$f_{GPP}$ values are close to 1 (except for very low PAR/VPD values and with a maximum of about 4-6% increase) which can be explained by almost identical $V_{cmax}$ response at lower temperatures in Fig. 1. At high PAR values with higher temperatures (25-30°C) and low VPD values (panel d) we find that GPP increases by about 6-14% which can be directly explained by the new increased $V_{cmax}$ response in that temperature range as indicated in Fig. 1.

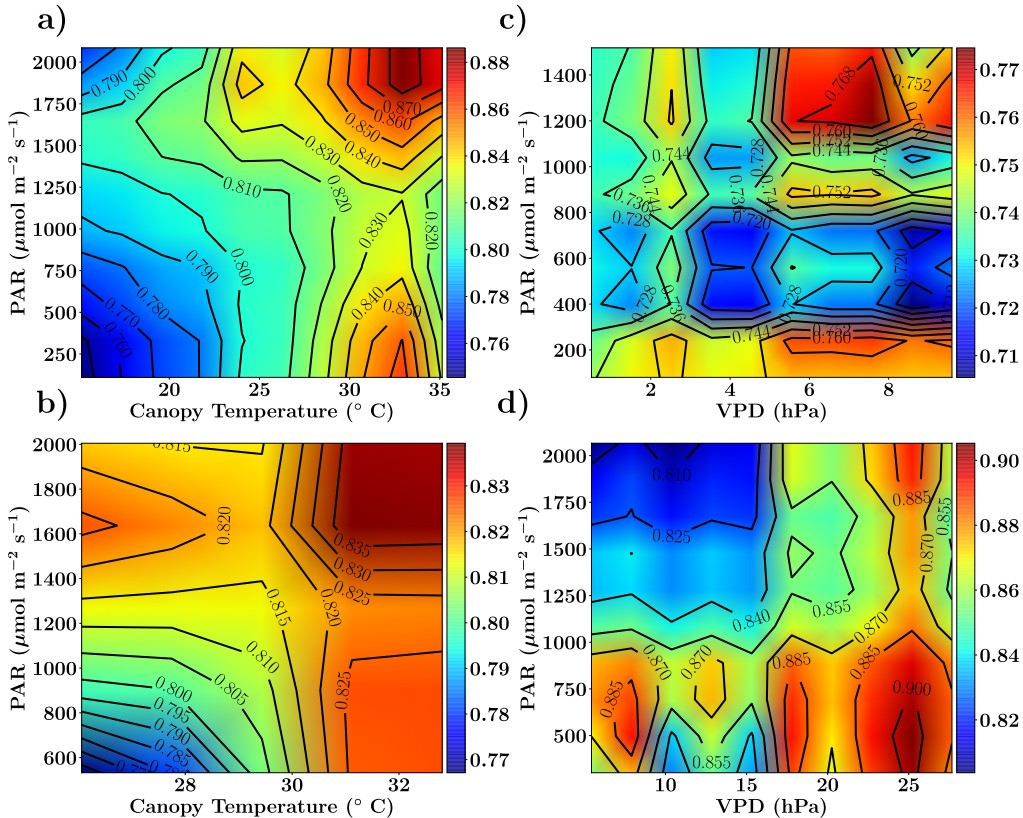

**Figure 4.** Figure showing the ratio of old and new SCOPE GPP ($f_{GPP} = \frac{GPP_{new}}{GPP_{old}}$) simulations as a function of PAR, canopy temperature and VPD for the C$_4$ species. The Nebraska Mead-1 site with corn growing for the year 2009 is used and the SCOPE simulations are driven with identical forcings and parameters for both the new and old simulations, the only difference being the implementation of photosynthesis and its temperature dependence (see text). The left column shows $f_{GPP}$ as a function of PAR and canopy temperature with data points in the low VPD range of 10-15 hPa (panel-a) and high VPD range of 25-30 hPa (panel-b). The right column shows $f_{GPP}$ as a function of PAR and VPD with data points in the low temperature range of 10-15 °C (panel-c) and high temperature range of 30-35 °C (panel-d).

5    Figure 4 shows similar plots and comparisons as described above in terms of $f_{GPP}$ but for corn (C$_4$) crop grown at the Mead-1 Ameriflux site in Nebraska Lincoln (see section 6.2.1 for some more site information). It appears from the results that there is only a decrease in canopy GPP for the new SCOPE both in the space of PAR-Canopy Temperature and PAR-VPD by about 5-30%, which can be attributed to photosynthesis being limited by light instead of rubisco at higher temperatures. As



mentioned previously, there is uncertainty associated with these contour diagrams although these are quite small ($< 10\%$) and it is found that these patterns are also nearly the same (not shown).

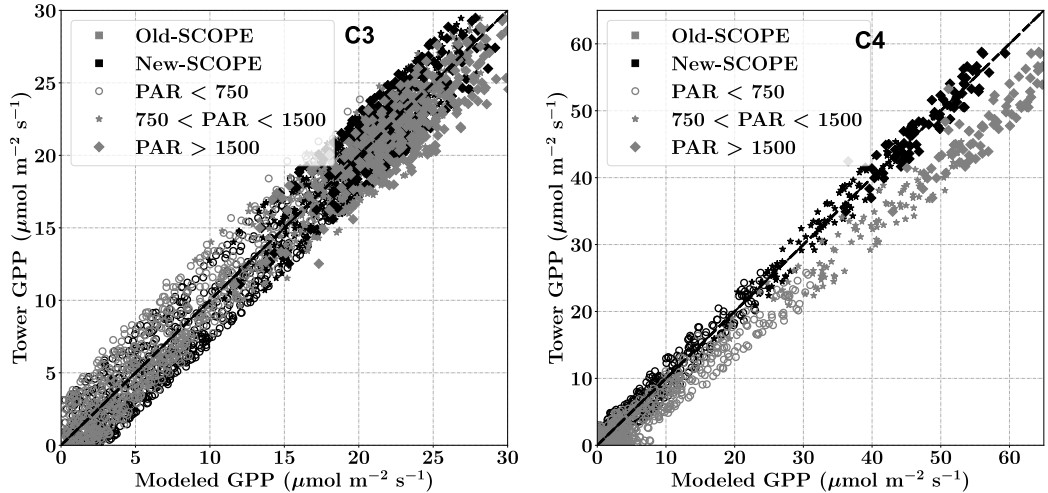

**Figure 5.** Figure showing the comparison of the tower observed vs modeled canopy GPP for new and old Photosynthesis implementations in the SCOPE model. The results are shown for both the C$_3$ (left) and C$_4$ (right) species. The model simulations for the C$_3$ species are for the Missouri Ozark Site comprising of Deciduous Broadleaf Forests, the C$_4$ simulations are for the Nebraska Mead-1 site with corn growing both for the year 2009.

Figure 5 shows the modeled vs observed comparison for new (as per CLM4.5) and old (previous) implementation of photosynthesis in SCOPE for both the C$_3$ and C$_4$ species. We select a set of points (time indices) from the new SCOPE simulations

5   at various PAR ranges whose modeled GPP fluxes are very close to the tower observed values. The points corresponding to the same time indices from the old SCOPE simulations are then selected for comparing the modeled GPP. It is found that for C$_3$ species there is not much change but for C$_4$ species the new SCOPE GPP values are generally reduced. This result is also in agreement with the contour plots as presented earlier. Overall, we find that the new model implementation of photosynthesis and its temperature dependence as well as $A - C_i$ iterations works well and only result in moderate, yet noticeable changes.

10  It also underlines that tabulated model parameters can only be optimized for a specific model implementation, which is not necessarily universally transferable to other carbon cycle models.



## 3 Formulation of Inverse Problem

The problem of ecosystem flux computation (e.g. GPP, Latent Energy, etc) from meteorological variables (e.g. VPD, air temperature, relative humidity etc) and other ecosystem parameters can be represented as:

$$Y = \mathcal{F}(X') + \epsilon \tag{4}$$

where, $\mathcal{F}() : X' \longrightarrow Y$ is a functional representation of the model, which maps the model input and parameter space $(X')$ quantitatively to the space of ecosystem fluxes $(Y)$, and $\epsilon$ represents the residual error which includes the precision error, the model error and random errors. In our case, SCOPE represents the forward model $\mathcal{F}()$, which is complex and moderately non-linear, representing a range of physics and canopy physiological processes. We can further represent our forward problem as:

$$Y = \mathcal{F}(X; p) + \epsilon \tag{5}$$

where $X$ represents the state vector of parameters to be retrieved, $p$ $(X, p \subset X'$ and $X' = X \cup p)$ is a vector of parameters which represents those quantities that influence the measurement, are known to some accuracy but not to be retrieved. We call these parameters the forward functional parameters. In our example $p$ represents the set of all fixed model (SCOPE) parameters not involved in the retrieval. The error term $\epsilon$ represents the measurement noise (e.g. noise or errors in the flux measurements).

Given a set of measurements Y, the optimal state vector $\hat{X}$ can be obtained by a generalized inverse method $\mathbf{R}$ represented as:

$$\hat{X} = \mathbf{R}(Y, \hat{p}, X_a, c), \tag{6}$$

where $\hat{p}$ represents the best estimate of the forward function parameters. The parameters $X_a$ and $c$ represents the parameters that do not appear in the forward function but they do affect the retrieval and are associated with uncertainties. $X_a$ represents the prior estimate of $X$ and $c$ represents any other parameters in the retrieval scheme as a catch-all for anything else that is

used in the retrieval method, which also includes the convergence criteria.

## 4 Linearization of the Forward Model

A basic prerequisite for inverting the forward model is to compute its sensitivity with respect to input parameters, i.e. the partial derivatives with respect to all the state vector elements (Jacobi matrix). For linear models, the Jacobians are independent of the actual state. In our case, the SCOPE forward model is moderately non-linear and its Jacobians need to be computed numerically

as analytical methods are currently lacking and hard to implement given some peculiarities in the FvCB equations.




With the Jacobian matrix and a simple forward model call, we can thus write a first order Taylor expansion for the forward model

$$F(X;p) = F(X;p)_{X=X_l} + \left.\frac{\delta F}{\delta X}\right|_{X=X_l} (X - X_l),$$ (7)

where $X_l$ is an arbitrary linearization point, $\frac{\delta F}{\delta X}$ is the partial derivative or Jacobian at the point $X = X_l$.

## 5   Iterative Retrieval Algorithm Setup

In the remainder of the paper, we will omit the vector of forward model parameters $p$ which are not a part of the retrieval framework. For the non-linear problem we use the maximum a-posteriori approach. The Bayesian solution for the non-linear inverse problem where the forward model is a general function of the state, the measurement error is Gaussian ($S_\epsilon$) and with a prior estimate of the state ($X_a$) with a Gaussian uncertainty in the prior state ($S_a$) (Rodgers, 2000) can be represented as:

$$-2\ln P(X|Y) = [Y - F(X)]^T S_\epsilon^{-1}[Y - F(X)] + [X - X_a]^T S_a^{-1}[X - X_a] + c',$$ (8)

where $c'$ is a constant. Our aim is to find the best estimate of the state vector $\hat{X}$ (denoted as $X$ henceforth) and an error characterization that describes the posterior *pdf*. The Gauss-Newton iteration steps for determining the state vector is given by:

$$X_{i+1} = X_i + (S_a^{-1} + K_i^T S_\epsilon^{-1} K_i)^{-1}[K_i^T S_\epsilon^{-1}[Y - F(X_i)] - S_a^{-1}[X_i - X_a]]$$ (9)

A brief derivation of Eqn. 9 is presented in appendix C, for a more in-depth treatment the reader is referred to (Rodgers, 2000).

### 5.1   Levenberg Marquardt Method

In general, the Gauss-Newton iterations discussed previously finds the minimum in one step if the cost function is quadratic with respect to $X$. However, in our case the cost function is not perfectly quadratic and the initial guess potentially far away from the solution, thus requiring multiple iterations. In addition, the non-linearity of the problem sometimes results in steps that would actually increase rather than decrease the fit quality. In order to overcome this issue Levenberg (1944) (Levenberg, 1944) and Marquardt (1963) (Marquardt, 1963) proposed the following iteration for non-linear least squares problem:

$$X_{i+1} = X_i + (KK^T + \gamma_i D)^{-1} K^T [Y - F(X_i)]$$ (10)

where, $\gamma_i$ is chosen at each step to minimize the cost function and $D$ is a diagonal scaling matrix to scale the elements of the state vector. It can be noted that for $\gamma_i \to 0$, leads to a Gauss-Newton iteration step and for $\gamma_i \to \infty$ tends to steepest descent

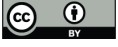

and further the step size tends to 0. It is also expected that the cost function will decrease corresponding to the decrease in $\gamma_i$ from infinity to zero. The value of $\gamma_i$ is sequentially updated at each iteration by evaluating the change in cost function. Here, we follow the general recommendations as outlined in (Marquardt, 1963; Rodgers, 2000).

The guidance for choosing the scaling matrix $D$ is that it must be positive definite. For the current problem we choose it to

5 be $S_a^{-1}$ (as in (Rodgers, 2000)) and apply the Levenberg Marquardt (LM) modification to the Gauss-Newton method (iteration equation C8), resulting in the following iterative inversion scheme:

$$X_{i+1} = X_i + [(1+\gamma)S_a^{-1} + K_i^T S_\epsilon^{-1} K_i]^{-1}\{K_i^T S_\epsilon^{-1}[Y - F(X_i)] - S_a^{-1}[X_i - X_a]\} \tag{11}$$

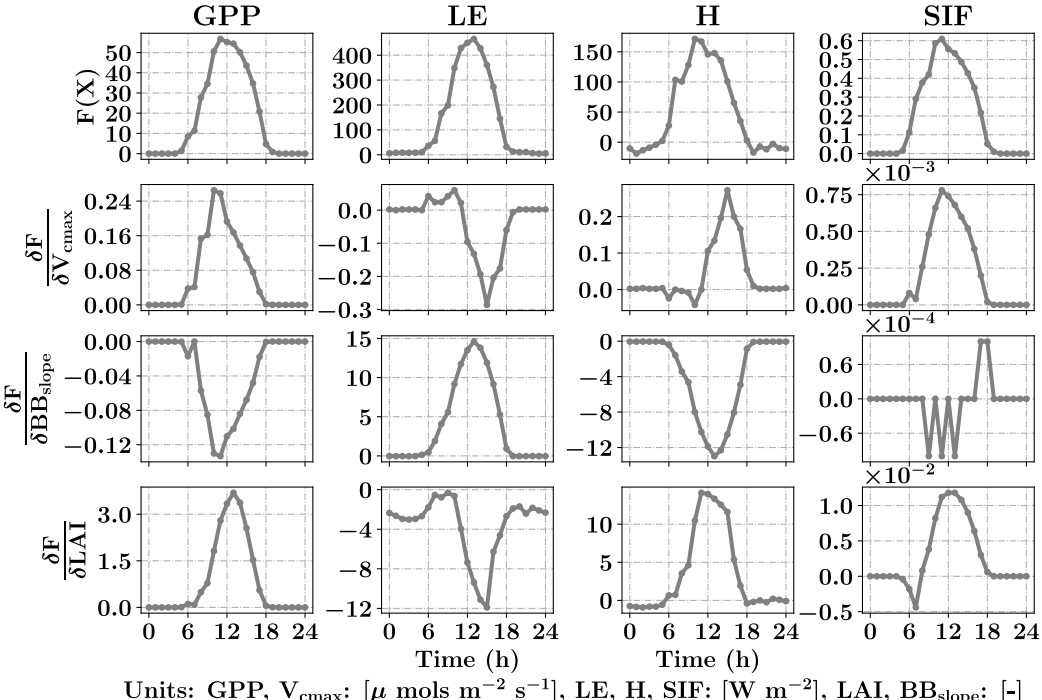

Units: GPP, $V_{\text{cmax}}$: [$\mu$ mols m$^{-2}$ s$^{-1}$], LE, H, SIF: [W m$^{-2}$], LAI, BB$_{\text{slope}}$: [-]

**Figure 6.** Diurnal variability of GPP, LE, H and SIF from SCOPE model simulations (top row) for a typical day in the growing season (August 3, 2010) for C$_4$ corn using data from the Nebraska Mead-1 flux tower site with parameter values $V_{\text{cmax}} = 50\ \mu$ mols m$^{-2}$ s$^{-1}$, BB$_{\text{slope}} = 7$ and LAI = 4. Second, third and fourth rows from the top shows the diurnal variability in the gradient of GPP, LE, H and SIF with respect to the parameters using SCOPE with positive perturbations $\delta V_{\text{cmax}} = 5\ \mu$ mols m$^{-2}$ s$^{-1}$, $\delta$BB$_{\text{slope}} = 1$ and $\delta$LAI = 0.5 which constitutes the Jacobian matrix for the inversions. It can be observed that the Jacobian matrix is non-linear with maximum values near the mid-day period. Our retrieval framework uses concatenated 3-day GPP and LE fluxes (modeled and observed) and their gradients successively within a 3-day window.



## 5.2 A Moving Window Set up of the Inversion Problem Using Flux Tower Observations

Figure 6 top row shows the SCOPE model simulations of GPP, LE, H and SIF for one day (August 3, 2010) in the growing season for $C_4$ corn using data from the Nebraska Mead-1 flux tower site with parameter values $V_{cmax} = 50 \mu$ mols m$^{-2}$ s$^{-1}$, $BB_{slope} = 7$ and LAI = 4. The second, third and fourth rows from the top shows the numerically computed partial derivatives

of GPP, LE, H and SIF with respect to the parameters using SCOPE with positive perturbations $\delta V_{cmax} = 5 \mu$ mols m$^{-2}$ s$^{-1}$, $\delta BB_{slope} = 1$ and $\delta$LAI = 0.5. Each column of figure 6 represents a row of Jacobian matrix used for the inversions. The figure clearly demonstrates the influence of each of the parameter variables in the state vector $(X)$ on the modeled fluxes $(F(X))$. We can observe the counteracting nature of variables and the fluxes from the Jacobian. For example, $V_{cmax}$ has positive gradient for GPP but negative for LE and likewise for GPP fluxes, $V_{cmax}$ has positive gradients but $BB_{slope}$ has negative gradient. It can

be noted that the nature of these gradients may vary depending on environmental conditions, such as incoming PAR as well as air temperature and vapor pressure deficit. This also creates diversity in the Jacobians over the diurnal cycle, which allows us to derive more than 2 parameters from 2 sets of measurements (GPP and LE). In Figure 6, we have not only shown derivatives of GPP and LE but also H and SIF (not used here). In this manuscript, we outline the general framework of parameter inversion, which can easily be modified to make use of more measurements such as H, SIF, reflectance or thermal emissions, all of which

can be modeled with SCOPE.

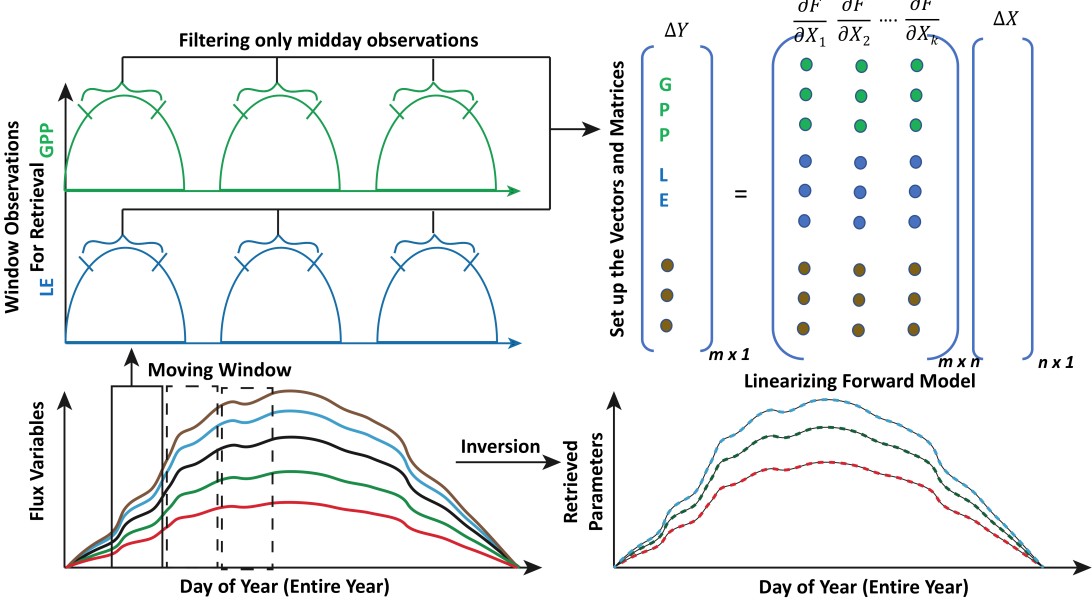

**Figure 7.** Illustration of moving window inversion retrieval setup. The bottom left part illustrates the ecosystem time series flux variables used for driving the SCOPE model. A $n$-day time window is selected for each retrieval in the yearly growing season and a time filter is implemented for concatenating the measurement vector in the retrieval windows. The top right shows the vector and matrix setup and the linearization of the forward model. The bottom right shows the retrieved model parameters implementing the moving window approach.





For setting up the observation vector $Y_{m \times 1}$ (see Eqn. 11), we use observations of carbon and latent energy fluxes from eddy covariance tower time series records. The observational error matrix ($S_{\epsilon[m \times m]}$) is assumed to be a diagonal matrix and computed using noise standard deviation as 10% of the observations. We assume an initial prior state value of the state vector ($X_{a[n \times 1]}$) as well as the prior error covariance matrix ($S_{a[n \times n]}$). As mentioned previously, the Jacobian matrix $K_{m \times n}$ is

computed numerically by a small perturbation to the value of the state vector $X_i + \delta$ (see Fig. 6) at a particular iteration step. The observed ($Y$) and modeled ($F(X)$) fluxes in the inversion framework are set up as a long concatenated vector as shown in Fig 7. The concatenation of different flux variables are done using a time filter to represent the part of the day we wish to include in the retrieval framework as illustrated in Fig. 7. This is logical as we have already demonstrated in Fig. 6 that the gradients are variable throughout the day. Ideally, the time filter applied for concatenating the data should capture the maxima

and a range of variations in the gradients, but at the same time reduce the data points to make the retrieval computationally efficient and further tend towards providing stable solutions (retrievals) of the parameter values. Further, the time filter helps to eliminate the night time anomalies in the observations for accurate parameter estimation. The assumptions behind the long term (seasonal) retrieval of important ecosystem and plant physiological parameters is that these parameters change significantly over the growing season but at a slower rate compared to and in response to the environmental and meteorological forcing.

Thus, the ecosystem parameters can be assumed to be constant over some finite time window. We implement this assumption to set up our inverse parameter retrieval framework for finite $n$-day contiguous moving windows over the entire growing season (Fig 7). We extend the one-day diurnal set up of $Y$, $F(X)$ and $K$ as shown in Fig. 6 to multiple days for setting up the $n$-day windows as illustrated by color coding in Fig. 7. After computing the necessary vectors and matrices for the $n$-day window, iterations are performed by applying the LM algorithm until convergence to obtain the posterior estimation of the state vector.

The retrieval window is moved over to the contiguous next $n$-days and the process is repeated. The retrieval thus proceeds for the entire length of the growing season (Fig 7). For our retrieval example, we choose a 3-day moving window which seems optimal for the plant response in terms of the photosynthesis parameters towards the change in environmental drivers.

## 5.3    Error Characterization and Convergence Criteria for the Retrievals

As mentioned in section 5.1, we have selected a convergence criteria for the parameter retrievals in each of the moving windows

based on the ratio of the true error to the expected error for each of the iteration steps. For each iteration step, the error (mismatch between observations and modeled values) ($\chi^2$) can be represented as:

$$\chi_i^2 = [Y - F(X_i)]^T S_\epsilon^{-1} [Y - F(X_i)] \tag{12}$$

The expected reduction in the error ($\chi_E^2$) which is computed after the inversion retrieval step and an expected value of the new state vector and without making update to either $X$ or $\gamma$ can be represented as:

$$\Delta\chi_{E_i}^2 = [Y - F(X_i) - K(X_{i+1}^* - X_i)]^T S_\epsilon^{-1} [Y - F(X_i) - K(X_{i+1}^* - X_i)] \tag{13}$$





The true reduction in the error ($\chi^2_T$) which is computed using a forward model run after the inversion retrieval step and using the updated value of state vector without actually making the update to either $X$ or $\gamma$ can be represented as:

$$\Delta\chi^2_{T_i} = [Y - F(X^*_{i+1})]^T S_\epsilon^{-1} [Y - F(X^*_{i+1})] \tag{14}$$

Finally, the ratio ($R$) of the true to expected change in error reads:

$$R = \frac{\chi^2_i - \Delta\chi^2_{E_i}}{\chi^2_i - \Delta\chi^2_{T_i}} \tag{15}$$

A few additional filters are implemented to ensure that the updated state vector is always within a physically meaningful space (e.g. $R < 0$ indicates negative $X$ and hence the state vector is not updated instead the $\gamma$ value is increased).

After convergence, the posterior error covariance matrix for the retrieved state vector $\hat{X}$ can be computed as:

$$S = [S_a^{-1} + K_i^T S_\epsilon^{-1} K_i]^{-1} \tag{16}$$

The reduction in error is defined as:

$$\zeta_i = 1 - \left(\frac{S_{jj}}{S_{a_{jj}}}\right)^{0.5} \tag{17}$$

In our LM retrieval process, we use the retrieved state vector $\hat{X}$ of the previous window as first guess (but not prior) for the current window. This saves computational cost and is based on the assumption that our state vector varies smoothly in time.

## 6 Results for Implementing the Inversion Framework in SCOPE

In this section, we discuss the results of optimal parameter estimation by applying the Bayesian inversion framework to three different ecosystems. The aim is to demonstrate the applicability for the retrieval of canopy and photosynthesis parameters using carbon and water fluxes, to demonstrate the seasonality of retrieved parameters and to further compare and contrast the results across the sites.

### 6.1 Data Filtering Criteria in the Moving Window Retrievals

Apart from the overall algorithmic steps as described previously, we apply the following filter criteria on the results and the data for a computationally efficient retrieval.

1. In constructing the observation vector $Y$ we apply a time of the day filter (e.g. data between 9 am and 4 pm and so on) for the initial forward SCOPE model.

2. For computing the Jacobians, a PAR based threshold (PAR $> 100\,\mu$ mols m$^{-2}$ s$^{-1}$) to ensure sensitivity of the measurement vector with respect to state vector variations and to minimize the occurrence of unreasonable fluxtower data (high fractional errors).

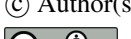



3. A filter is implemented to check and ensure that the state vector remains positive at every iteration. If somehow due to a small enough $\gamma$ the state vector is negative, the $\gamma$ value is adjusted in an iterative manner to keep it within bounds.

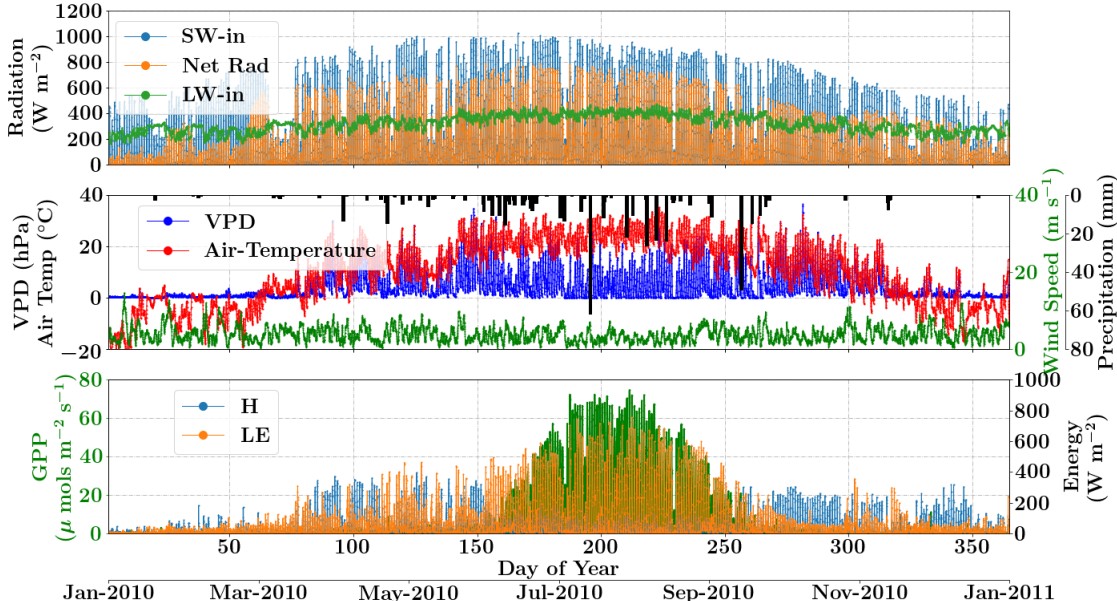

**Figure 8.** Figure showing the diurnal and seasonal variability of important environmental and meteorological forcings together with the tower observed fluxes of carbon and energy used in SCOPE model inversions for the Nebraska Mead flux tower site. The variables in the top and middle panels are used as inputs to the SCOPE model and the variables in bottom panel is used as a target in a moving window retrieval approach.

## 6.2 Retrieval Results for the Nebraska Mead-1 site

### 6.2.1 Site Description

5  The Nebraska Mead-1 site is a part of the Ameriflux network located in Lincoln, Nebraska and is one of the three cropland sites at the University of Nebraska Agricultural Research and Development Center. The site has continuous data records from 2001 till present (Suyker et al., 2005). This site is a continuously irrigated corn ($C_4$ species) crop site, with mean annual precipitation of 790 mm and mean annual temperature of 10.07 °C. We choose the year 2010 and an hourly time resolution for the analysis. The site meteorology and forcing variables relevant to the SCOPE inversion retrievals are shown in Figure 8. The top two

10  panels show the environmental forcing variables which are used as input (except precipitation) in the SCOPE simulations. The bottom panel represents carbon (GPP) and energy (LE, H) fluxes, which are used to construct the observation vector $Y$. We can clearly see that the growing season extends from around June through September, coinciding with high temperature, VPD and net radiation. We focus on the retrieval of the parameters $V_{cmax}$, $BB_{slope}$ and LAI during this entire growing season.



| | Prior State Vector | | | Prior Error ($\sigma$) | | | |
|---|---|---|---|---|---|---|---|
| **Site** | $V_{cmax}$ | $BB_{slope}$ | LAI | $V_{cmax}$ | $BB_{slope}$ | LAI | Duration (hrs) |
| Mead-1 ($C_4$) | 50 | 4 | 7 | 30 | 5 | 1 | 9 - 16 |
| Missouri Ozark ($C_3$) | 80 | 4 | 2 | 20 | 5 | 1 | 10 - 14 |
| Niwot Ridge ($C_3$) | 80 | 4 | 3.8 | 20 | 5 | $10^{-5}$ | 9 - 16 |

**Table 1.** Prior Values for LM Inversion (Units: $V_{cmax}$: [$\mu$ mols m$^{-2}$ s$^{-1}$], $BB_{slope}$, LAI: [-]

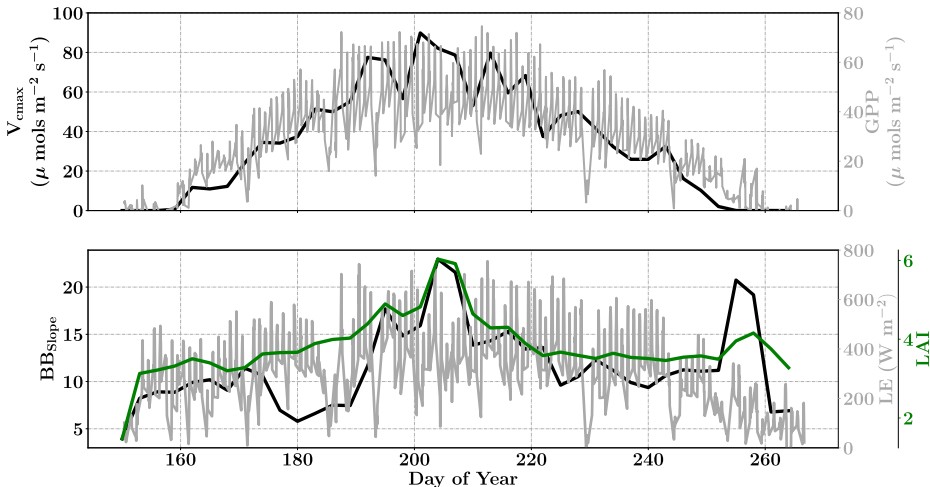

**Figure 9.** Figure showing the seasonal variability in retrieved parameter values of $V_{cmax}$, $BB_{slope}$ and LAI for the Nebraska Mead-1 site using a 3-day moving window inversion approach for the year 2010. The actual points in the time series (grey lines) of the GPP and LE fluxes used as the target observations (Y) for the moving window inversion approach are shown in the background. The results show reasonable trends in the retrieved parameters along with their sensitivity to GPP and LE fluxes across the growing season.

### 6.2.2 Inversion Parameters and Results

For each of the retrieval windows, the prior value of the state vector along with prior errors and day time duration, which is used for filtering the GPP and LE observations are shown in Table 1. Here we use a purely diagonal prior error covariance matrix, with zero off-diagonal elements. Figure 9 shows the retrievals of parameters $V_{cmax}$, $BB_{slope}$ and LAI. The grey time series

5 of GPP and LE values in the background are the actual filtered values used for constructing the observation (Y) and modeled ($F(X)$) vectors corresponding to each retrieval window. We find a seasonal variability in the parameters, which follow a similar pattern in GPP or LE. The retrieved $V_{cmax}$ shows a very strong seasonality with GPP. There is also a considerable seasonality in $BB_{slope}$ and LAI. The values during the growing season for the corn crop are found to be reasonable and realistic for all parameters. The LAI increases steadily from 2 to about 6 and then declines gradually. As expected, the optimized parameters

10 are quite sensitive to the variation in GPP and LE, for example around DOY 190, where there is sudden dip in the fluxes. The





large variability in $BB_{slope}$ and LAI around DOY 200 and DOY 260 may be partially attributed to the largest rainfall events (see Fig. 8). Part of the variability and correlation between $BB_{slope}$ and LAI may be due to the diminishing role of soil evaporation (parameterized by a single resistance in SCOPE) with increasing LAI. Another part may be due to evaporation from the wet canopies which is not currently represented in SCOPE. This may cause the inversion to overestimate $BB_{slope}$, but it may not

represent the gas exchange between the stomata and the leaf surface.

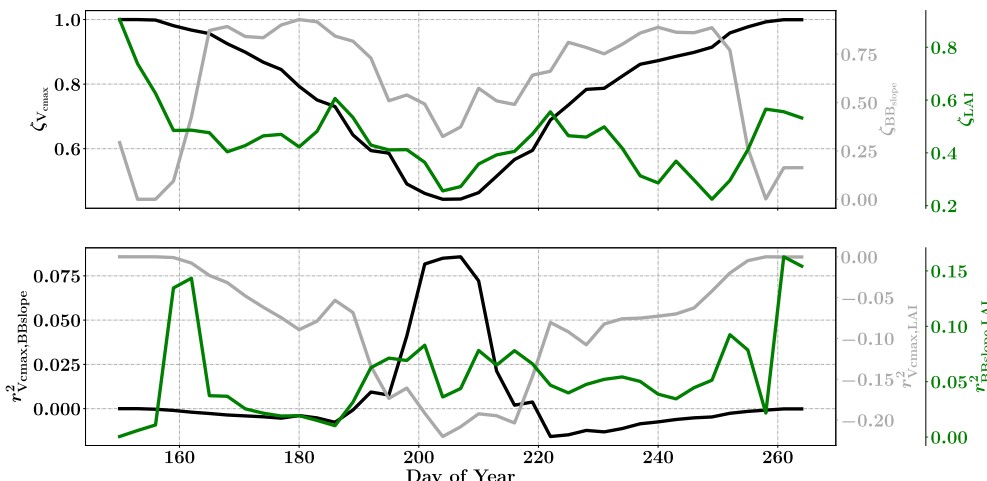

**Figure 10.** Figure showing the seasonal variability of the posterior error reduction ($\zeta$) and correlation coefficient of the retrieved parameter values of $V_{cmax}$, $BB_{slope}$ and LAI for the Nebraska Mead -1 site using a 3-day moving window inversion approach for the year 2010. The top panel shows the $\zeta_{V_{cmax}}$, $\zeta_{BB_{slope}}$ and $\zeta_{LAI}$ for the entire growing season and the bottom panels shows the correlation coefficients (normalized off-diagonal elements of posterior error covariance matrix) among these variables. Both the $\zeta$ and correlation coefficients are computed using the final Jacobian matrix at the end of each retrieval window.

Figure 10 (top panel) shows the final posterior error reduction ($\zeta_i$-eqn. 17) of the retrieval iterations for each moving window. The value of $\zeta_i$ is computed from the diagonal elements of the posterior error covariance matrix. We find a significant reduction in the posterior errors of the variables in the state vector. There is a strong seasonality in $\zeta_{V_{cmax}}$ and $\zeta_{BB_{slope}}$ values and moderate to none for the $\zeta_{LAI}$. The posterior error covariance matrix also indicates whether the retrieved parameters are truly

independent (as in the case of a diagonal matrix) or whether they co-vary (indicated by significant off-diagonal elements). The error correlation is given by $r^2_{x,y} = \frac{COV(x,y)}{\sigma_x \sigma_y}$, it can be either positive or negative indicating whether the parameters move in the same or opposite direction. In general, high correlations indicate that we cannot fit these parameters independently. Figure 10 (bottom panel) shows the growing season error correlation patterns between the three parameters from the retrievals.From the results, it can be seen that during the peak growing season $r^2_{BB_{slope},LAI}$ is positive indicating they are changing in sync and

in comparison $r^2_{V_{cmax},LAI}$ is negative indicating the counteracting influence of these variable pairs. However, $r^2_{V_{cmax},BB_{slope}}$ is positive until DOY 220 thereafter it is negative indicating both favorable and competing effects during the different parts of



the growing season. The error correlations $r^2_{\mathrm{BB_{slope},LAI}}$ and $r^2_{\mathrm{V_{cmax},LAI}}$ are stronger than $r^2_{\mathrm{V_{cmax},BB_{slope}}}$. It is also found that the error correlations increase during the peak growing season.

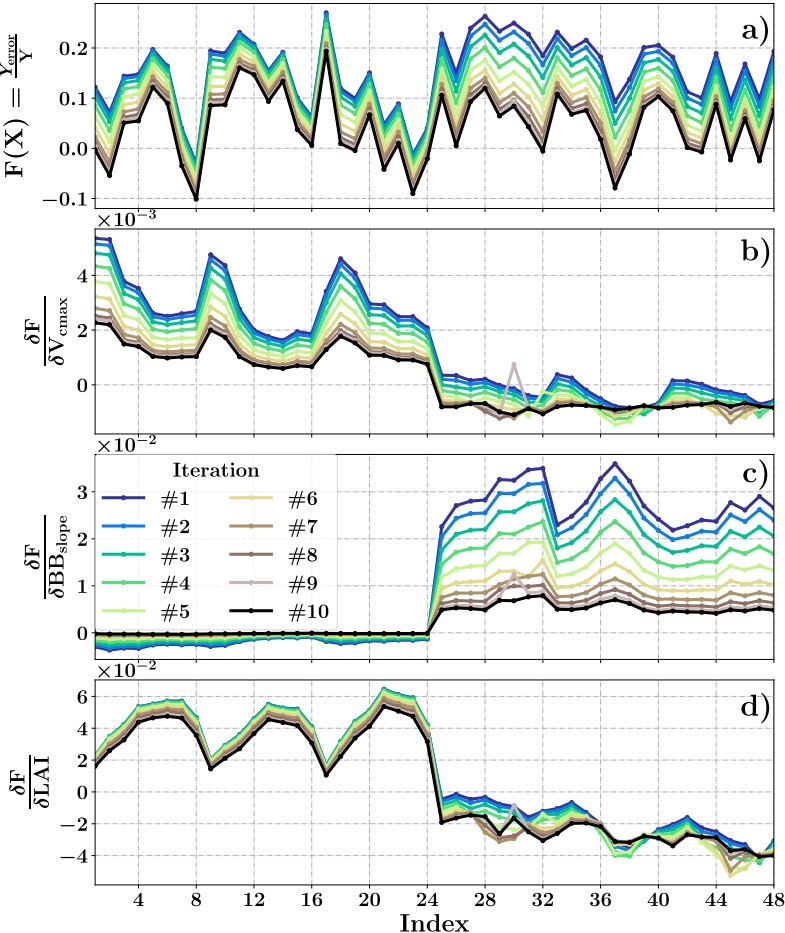

**Figure 11.** Figure showing the evolution of the Jacobian Matrix for one retrieval window(DOY 213-216) for the Nebraska Mead-1 site in 2010. The normalized mismatch $Y_{error}$ between the observation and modeled vector (observed minus modeled) for the 3-day window composed of GPP (indices 1-24) and LE (indices 25-48) time series concatenated together for each retrieval iteration is shown in panel (a). The normalized gradients of the forward model (SCOPE) with respect to the variables in the state vector after each update step of the LM algorithm are shown in the panels (b), (c) and (d) respectively. The gradient decreases with each iteration and the observations of GPP are weighed more for $V_{cmax}$, Observations of LE more for $BB_{slope}$ and both for the LAI retrievals.

Figure 11 shows the evolution of the Jacobian matrix for a typical retrieval window represented by DOY 213-216. The lines in the top panel (a) represent the concatenated normalized errors between the observation and modeled vector. It is found that

5     the errors decrease monotonically with each retrieval iteration. The panels (b), (c) and (d) (Fig. 11) represent the normalized





gradients with respect to each variable in the state vector corresponding to each iteration step of the LM inversion approach in the retrieval window. The indices 1-24 on the $x$-axis represents the concatenated and filtered 3-day GPP and indices 25-48 represents the LE. We find that the gradients are not constant, decrease with each iteration step and has a somewhat diurnal structure to it, indicating the non-linear nature of the problem. Further, the influence of gradient values are higher in terms

5  of $V_{cmax}$ for GPP and $BB_{slope}$ for LE and the values are very small for the vice-versa cases (also see Fig. 6). LAI appears to have interesting competing behavior with positive gradients for GPP and slightly negative for LAI although it varies with different iterations. Finally, figure 12 represents the net improvement in canopy GPP and LE fluxes due to the optimized state vector over their prior values. The first column represents the diurnal and seasonal variability in the time series of GPP and LE fluxes with optimized and unoptimized parameters and further its comparisons with flux tower values. The right column

10  represents the one-to-one comparisons of the same. We find a significant improvement in the estimation of GPP ($R^2 = 0.94$) and the optimized parameters are able to capture the growing seasonal variability well as measured by flux tower observations (slope = 1.04).

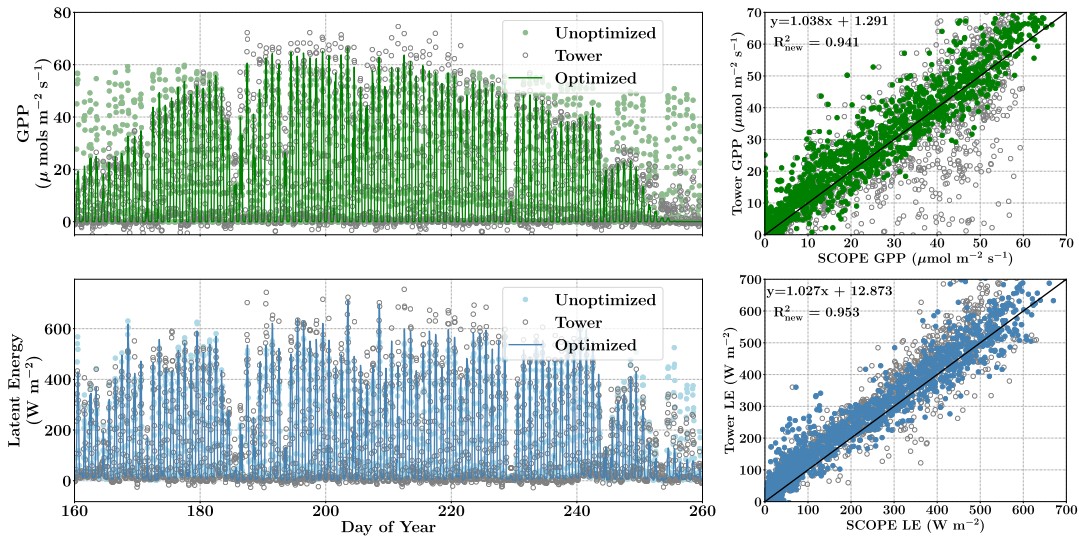

**Figure 12.** Figure showing the improvement in diurnal and seasonal variability in modeling the GPP and LE fluxes with optimized parameters over prior values using SCOPE for the Nebraska Mead-1 site for the year 2010. The figure also shows the one-to-one comparison (indicated by black-line) with the observed flux tower values. The optimization of the photosynthetic parameters improves the accuracy of computing the carbon and water fluxes as indicated by the $R^2$ value and the equation of the regression line.

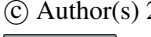



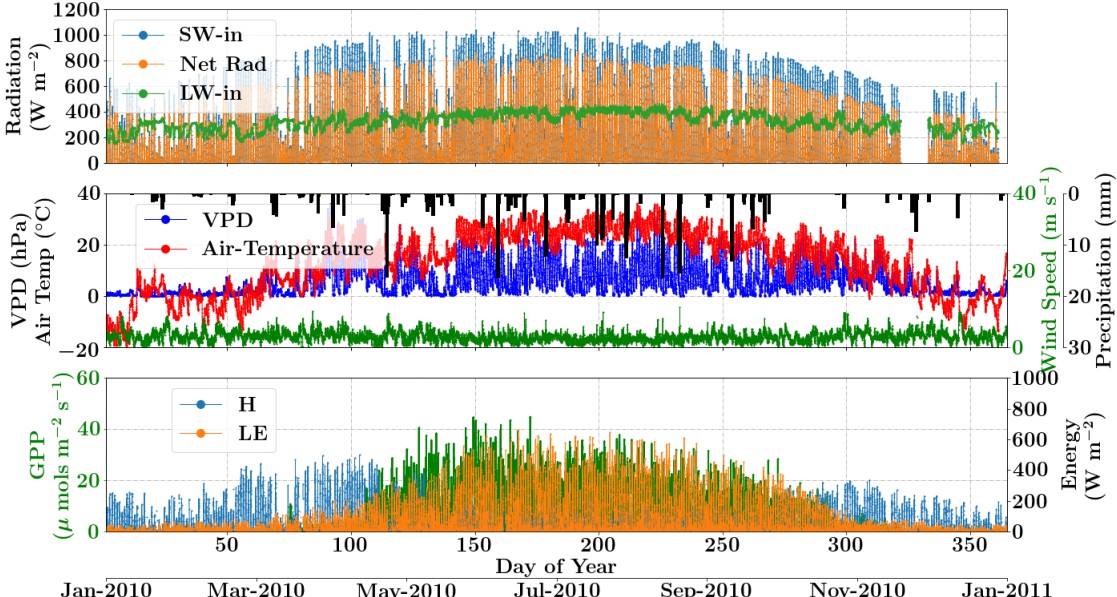

**Figure 13.** Figure showing the diurnal and seasonal variability of important environmental and meteorological forcings together with the tower observed fluxes of carbon and energy used in SCOPE model inversions for the Missouri Ozark flux tower site. The variables in the top and middle panels are used as inputs to the SCOPE model and the variables in bottom panel is used as a target in a moving window retrieval approach.

## 6.3 Retrieval Results for the Missouri Ozark site

### 6.3.1 Site Description

The Missouri Ozark site is also a part of the Ameriflux network located in the University of Missouri Baskett Wildlife Research area, situated in the Ozark region of central Missouri. It is uniquely located in the ecologically important transitional zone

5 between the central hardwood region and the central grassland region of the US (Gu et al., 2006). This site has a mean annual precipitation of 986 mm and a mean annual temperature of 12.11 °C and has continuous data record from 2004 till present. It is a deciduous broadleaf forest site comprising of $C_3$ plant species. We use the half hourly time resolution datasets from the year 2010 in the present analysis. The site meteorology and forcing variables relevant to the SCOPE inversion retrievals for the year 2010 are shown in Figure 13. The top two panels show the environmental forcing variables which (except precipitation)

10 are used as input in the SCOPE model simulations. The bottom panel represents the observations of carbon (GPP) and energy (LE, H) fluxes which are used to construct the observation vector $Y$ (composed of GPP and LE data concatenated together) for each of the windowed inversions. We find that the growing season is longer for this site from around March, when the air temperatures starts to become positive and slightly warmer, till around November when the trees lose leaves and there is onset





of the fall season. Site temperatures are quite high with high VPD around June and July. Again, we focus on the retrieval of the parameters $V_{cmax}$, $BB_{slope}$ and LAI during this entire longer growing season.

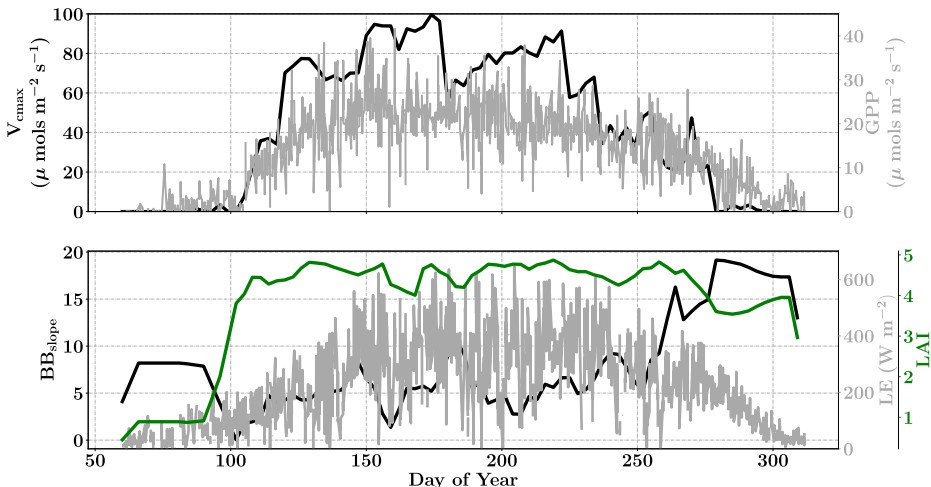

**Figure 14.** Figure showing the seasonal variability in retrieved parameter values of $V_{cmax}$, $BB_{slope}$ and LAI for the Missouri Ozark site using a 3-day moving window inversion approach for the year 2010. The actual points in the time series (grey lines) of the GPP and LE fluxes used as the target observations (Y) for the moving window inversion approach are shown in the background. The results show reasonable trends in the retrieved parameters along with their sensitivity to GPP and LE fluxes across the growing season.

### 6.3.2 Inversion Parameters and Results

The assumed prior value of the state vector and prior errors and day time duration which is used for filtering the GPP and LE observations for the retrieval windows are shown in Table 1. As for the Mead-1 site in the LM retrievals we have assumed the prior error covariance to be zero, along with the same assumptions for the initial guess of the state vector. Figure 14 shows the results for the retrieval of parameters $V_{cmax}$, $BB_{slope}$ and LAI. The grey time series of GPP and LE values in the background are the actual values used for constructing the observation vector $Y$ corresponding to each retrieval window for parameter retrieval. As indicated earlier the Ozark dataset is half hourly resolution therefore we have more number of observations to match the modeled fluxes in each of the retrieval windows. The retrieval for this site is also carried out over much longer duration covering almost the entire year.

Similar to the previous site, we find a strong seasonal variability in $V_{cmax}$, following the patterns in GPP and LE. We also find a steady increase in $BB_{slope}$ until it becomes constant with small fluctuations towards the middle of the year. The increasing trend in $BB_{slope}$ around DOYs 160, 175, 230 and 250 may all be associated with individual (comparatively large) rainfall events around these days (see Fig. 13). The LAI evolves from near zero rapidly to around 4-5 in the March-April time frame, which indicate rapid appearance of new leaves in the spring time. The LAI also further remains nearly constant for most part of the growing season from around DOY 100 to 300. This can be explained by the fact that after the leaves are





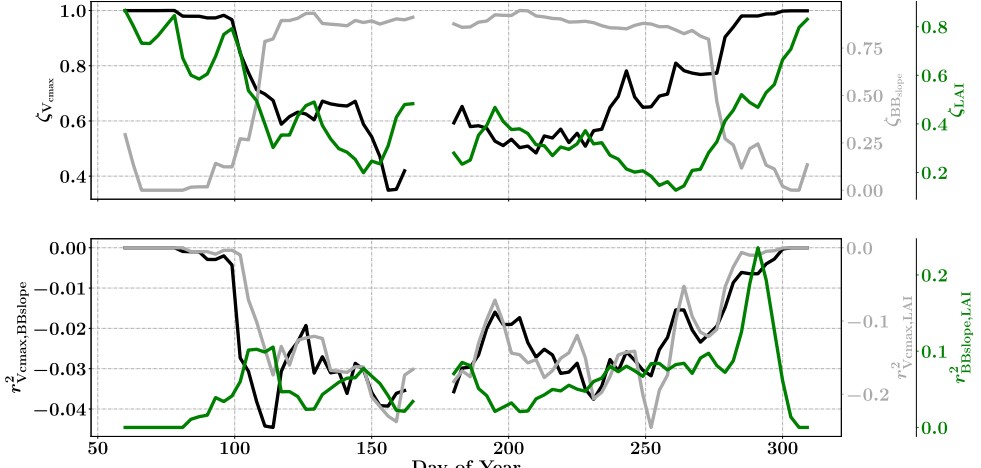

**Figure 15.** Figure showing the seasonal variability of the posterior error reduction ($\zeta$) and correlation coefficient of the retrieved parameter values of $V_{cmax}$, $BB_{slope}$ and LAI for the Missouri Ozark site using a 3-day moving window inversion approach for the year 2010. The top panel shows the $\zeta_{V_{cmax}}$, $\zeta_{BB_{slope}}$ and $\zeta_{LAI}$ for the entire growing season and the bottom panels shows the correlation coefficients (normalized off-diagonal elements of posterior error covariance matrix) among these variables. Both the $\zeta$ and correlation coefficients are computed using the final Jacobian matrix at the end of each retrieval window.

fully developed the LAI of the deciduous forest stand reaches its maximum value. It is observed that the inversion framework captures the seasonal variability in LAI.

Figure 15 shows the final posterior error reduction of the retrieval iterations for each moving window. It is found that there is significant reduction in the posterior errors for $BB_{slope}$ and $V_{cmax}$. We find that $\zeta_{V_{cmax}}$ and $\zeta_{BB_{slope}}$ has the same trend

and seasonality as that of retrieved $V_{cmax}$ and LAI respectively. The evolution of the error correlations is again interesting and very different from what we found for the Mead corn-$C_4$ case. The correlations $r^2_{V_{cmax},BB_{slope}}$ and $r^2_{V_{cmax},LAI}$ are both negative indicating counteracting effects of these variable pairs. In comparison $r^2_{BB_{slope},LAI}$ is positive indicating in-sync or similar behavior towards either positive or negative change. Both the error correlations $r^2_{V_{cmax},BB_{slope}}$ and $r^2_{V_{cmax},LAI}$ are high in the middle of the growing season with the later an order of magnitude higher, indicating the necessity for jointly optimizing

this parameter pair. The error in LAI is high after around DOY 250, and so is its correlation with $BB_{slope}$ indicating increased uncertainty for the period.

Finally, Figure 16 represents the net improvement in canopy GPP and LE fluxes due to the optimized state vector over their prior values. The left column represents the diurnal and seasonal variability in the time series of GPP and LE fluxes ($R^2 = 0.79$) with optimized and unoptimized parameters and further its comparisons with flux tower values. The right column represents

the one-to-one comparisons of the same. We find there is a significant improvement in the estimation of both GPP and LE fluxes with the optimized parameters. The optimized state vector is able to capture the growing seasonal variability well (slope of regression lines: GPP = 0.84 and LE = 0.98). The un-optimized prior values severely under-predict both fluxes. The optimal



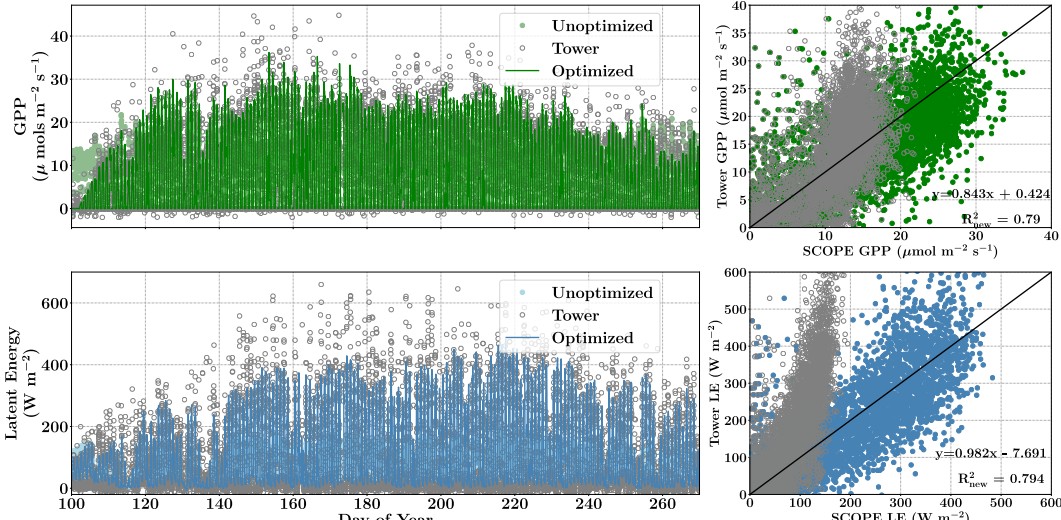

**Figure 16.** Figure showing the improvement in diurnal and seasonal variability in modeling the GPP and LE fluxes with optimized parameters over prior values using SCOPE for the Missouri Ozark site for the year 2010. The figure also shows the one-to-one comparison (indicated by black-line) with the observed flux tower values. The optimization of the photosynthetic parameters improves the accuracy of computing the carbon and water fluxes as indicated by the $R^2$ value and the equation of the regression line.

inversion method is able to capture the seasonal dynamics in the photosynthetic and canopy structural parameters for accurate prediction of the fluxes.

## 6.4 Retrieval Results for the Niwot Ridge Site

### 6.4.1 Site Description

5  The Niwot Ridge site is also a part of the Ameriflux network located in a subalpine forest ecosystem just below the continental divide near Nederland, Colorado. The average elevation of this site is 3050 m and is one of the high alpine evergreen needleleaf forests with $C_3$ plant species (Burns et al., 2016). This ecosystem is nearly 100 years old thus very different from the Mead and the Ozark sites (Monson et al., 2002). This site has a mean annual precipitation of 800 mm and a mean annual temperature of 1.5 °C and has continuous data record from 1998 till present. This site is thus the coldest and driest among the three.

10  Once again we choose the year 2010 for the current analysis and we have used dataset at an half hourly time resolution. The site meteorology and forcing variables relevant to the SCOPE inversion retrievals for the year 2010 are shown in Figure 17 which is very different from the previous two sites. The top two panels show the environmental forcing variables which (except precipitation) are used as input in the SCOPE model simulations. The bottom panel represents the observations of carbon (GPP)

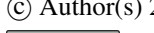



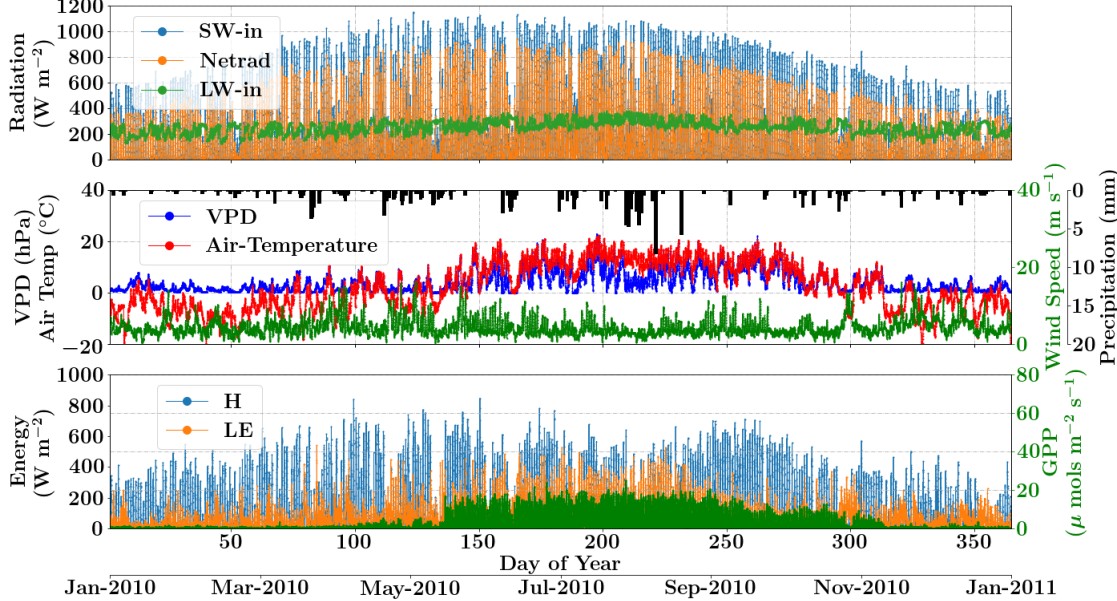

**Figure 17.** Figure showing the diurnal and seasonal variability of important environmental and meteorological forcings together with the tower observed fluxes of carbon and energy used in SCOPE model inversions for the Niwot Ridge flux tower site. The variables in the top and middle panels are used as inputs to the SCOPE model and the variables in bottom panel is used as a target in a moving window retrieval approach.

and energy (LE, H) fluxes which are used to construct the observation vector $Y$ (composed of GPP and LE data concatenated together) for each of the windowed inversions. The trees are evergreen and there is no growing season although we find that the photosynthetic activity and fluxes of GPP are increasing in the period between May and October. Although the magnitudes are much smaller compared to either the Mead or Ozark sites. The sensible heat at the site is also larger during this period

5    compared to the latent heat fluxes. We will focus on this period for the retrieval of the parameters $V_{cmax}$, $BB_{slope}$ and LAI from GPP and LE fluxes.

### 6.4.2   Inversion Parameters and Results

For the retrievals the prior value of the state vector along with prior errors and day time duration used in the retrieval windows are shown in table 1. Same as the earlier examples for the LM retrievals we have assumed the prior error covariance to be zero.

10   For this evergreen site, we have assumed a prior value of LAI equal to 3.8 from previous literature (Monson et al., 2009) and assumed the prior error on LAI to be very small. This way, we set the LAI values to remain constant for the retrieval windows in the inversion framework, which improves the retrieval of the other state vector parameters. In the future, other observables such as near-infrared reflectance could be used to add constraints on LAI, which will help decouple correlated errors.





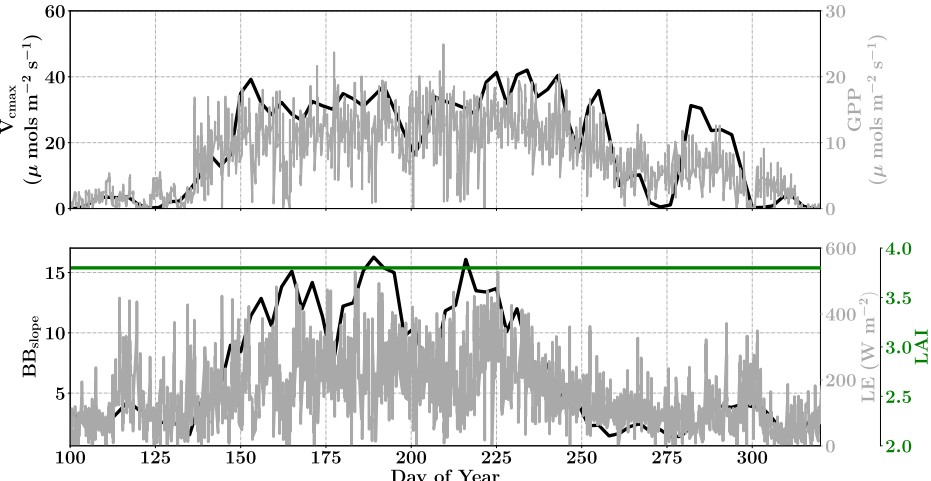

**Figure 18.** Figure showing the seasonal variability in retrieved parameter values of $V_{cmax}$, $BB_{slope}$ and LAI for the Niwot Ridge site using a 3-day moving window inversion approach for the year 2010. The actual points in the time series (grey lines) of the GPP and LE fluxes used as the target observations (Y) for the moving window inversion approach are shown in the background. The results show reasonable trends in the retrieved parameters along with their sensitivity to GPP and LE fluxes across the growing season.

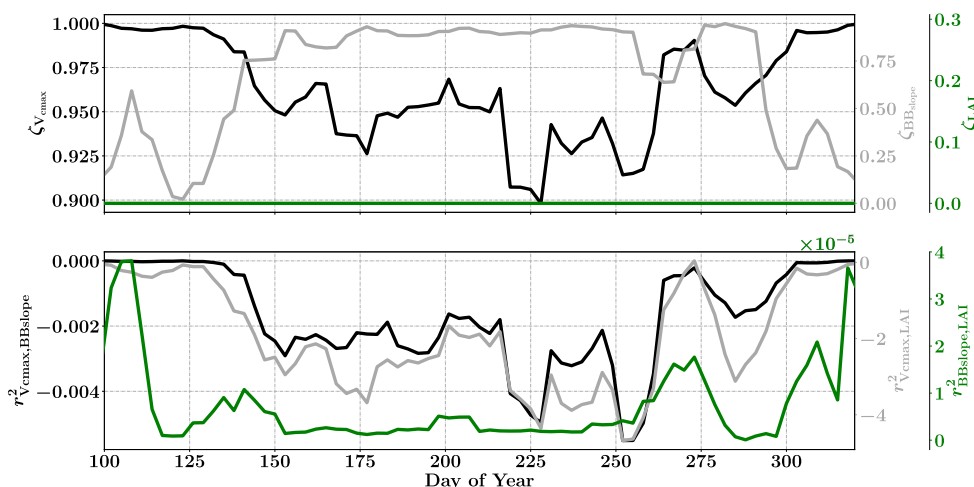

**Figure 19.** Figure showing the seasonal variability of the posterior error reduction ($\zeta$) and correlation coefficient of the retrieved parameter values of $V_{cmax}$, $BB_{slope}$ and LAI for the Niwot Ridge site using a 3-day moving window inversion approach for the year 2010. The top panel shows the $\zeta_{V_{cmax}}$, $\zeta_{BB_{slope}}$ and $\zeta_{LAI}$ for the entire growing season and the bottom panels shows the correlation coefficients (normalized off-diagonal elements of posterior error covariance matrix) among these variables. Both the $\zeta$ and correlation coefficients are computed using the final Jacobian matrix at the end of each retrieval window.





Figure 18 shows the results for the retrieval of parameters $V_{cmax}$, $BB_{slope}$ and LAI. The grey time series of GPP and LE in the background are the actual values used for constructing the observation vector $Y$ corresponding to each retrieval window for parameter retrieval. As indicated earlier the LAI value is maintained constant by setting a very low prior error in the inversion framework. The $V_{cmax}$ values again follow the similar trend as the GPP fluxes across the entire active season and the inversion

captures the rise and fall in the GPP trends extremely well. The slight dip and rise in the 190-210 day period follows the GPP observational data and may be attributed to consecutive cloudy days. The trends in $BB_{slope}$ seems to closely follow the variations in LE fluxes and captures the seasonality well.

Figure 19 shows the final posterior error reduction of the retrieval iterations for each moving window. It is found that the posterior error reduction for $BB_{slope}$ and $V_{cmax}$ are significantly high and almost zero for the LAI (as expected). We find

that $V_{cmax}$ has a similar trend in error reduction and correlation with both $BB_{slope}$ and LAI, although they are two orders of magnitude different. This is attributed to very low prior error on LAI used for this example. The evolution of the error correlations follows similar trend as in the previous Missouri Ozark $C_3$ site. We find $r^2_{V_{cmax},BB_{slope}}$ and $r^2_{V_{cmax},LAI}$ are both negative and $r^2_{BB_{slope},LAI}$ is positive. There is a sharp discontinuity around DOY 250-260 in terms of $\zeta_{BB_{slope}}$ and $r^2_{V_{cmax},LAI}$, $r^2_{V_{cmax},BB_{slope}}$ this is probably due to quality and/or discontinuity in the observational fluxes and environmental forcings.

## 7 Discussion and Conclusion

Our results demonstrate the feasibility of a moving window inversion approach for successful retrieval of key ecosystem parameters by constraining the SCOPE modeled carbon and water fluxes with eddy covariance flux tower observations. The moving window retrieval approach is a novel method for the retrieval of the seasonal parameter variability. The SCOPE model handles both the $C_3$ and $C_4$ photosynthetic pathways and could thus be applied to study a wide variety of ecosystems. We

demonstrate the approach here for climate and productivity gradients across agricultural, deciduous broadleaf forest and sub-alpine evergreen forest ecosystems.

There is strong evidence from measurements that under normal conditions LAI and photosynthetic parameters have seasonal variability (Wang et al., 2008; Wilson and Baldocchi, 2000; Wilson et al., 2000) which correlate with observations of energy fluxes. Our model inversion results are in alignment and agree well with these observations.

The developed Bayesian optimal inversion framework in SCOPE is flexible to incorporate other constraining fluxes and variables as well as other elements in the state vector to be optimized. SIF and visible to shortwave reflectance data are examples of such constraining observations, which are also obtained from SCOPE model simulations. Global time series of SIF observations, which provide a direct probe into photosynthetic machinery are becoming available from space based (Frankenberg et al., 2014; Guanter et al., 2014; Frankenberg et al., 2011) and ground based observations (Frankenberg et al.,

2016). Further, the retrieval framework could also be used to retrieve the photosynthetic temperature dependency parameters such as entropies and activation energies, which are even harder to measure directly but might be crucial, especially as the modeling of the ecosystem response to a warming climate mostly neglects potential changes in temperature dependencies of $V_{cmax}$. Our ongoing and future research efforts aims to address these questions by incorporating these newer observations





(Zhang et al., 2014) with carbon and water fluxes in the inversion framework. The Jacobians from our inversion results indicate that the optimal estimation is non-linear and therefore requires an iterative solution. Our Bayesian non-linear estimation allows us to compute accurate posterior uncertainty estimates.

It should be mentioned that our study focuses on the conceptual inversion framework, demonstrating a novel approach for
estimating important ecosystem parameters for modeling the dynamics of coupled carbon and water fluxes across ecosystems. However, there are opportunities for improving the overall inversion approach to better estimate the parameters. SCOPE allows us to ingest a variety of other observable to constrain the parameters space, including spectrally resolved reflectance (which can constrain LAI and chlorophyll content) as well as thermal emissions (which constrain LE) and SIF (which constrains APAR and $V_{cmax}$). In addition, in the current implementation the inversion approach may not optimize and retrieve the key parameters
well for ecosystems undergoing drought with limited soil water availability. An example is a typical mid-summer drought leading to a stomatal closure or productivity maximums being reached in the early morning hours which has a large phase difference with the diurnal PAR forcing. We hypothesize these may be due to some deficiencies in the process representation in SCOPE. There are competing optimality theories between whether the $BB_{slope}$ (Van der Tol et al., 2008b, a; MÄKELÄ et al., 1996) or $V_{cmax}$ (Xu and Baldocchi, 2003) is most affected by drought during growing season. An improvement in the current
framework could be better process representation of the soil moisture status in the stomatal conductance model within SCOPE either through implementation of leaf water potential (Tuzet et al., 2003) or optimality approach between water loss and carbon gain (Medlyn et al., 2011). Our inversion framework which jointly constrains both parameters may then be able to provide a better solution in drought conditions which is a subject for future investigation.

Our inversion framework is highly flexible in terms of allowing an arbitrary number of prior and retrieval parameters, which
could be tuned for better estimation of the key ecosystem parameters. The step-wise optimization approach within SCOPE also automatically weighs the carbon and water fluxes towards optimal state vector estimation without any predefined constraining measures (Wolf et al., 2006) towards particular parameters. The developed method also emphasizes the need to be able to compute the Jacobian matrices numerically for complex biophysical models, perhaps as an auxiliary but important output for general carbon cycle and terrestrial ecosystem modeling paradigms, which could facilitate inversions and error characterization
on a global scale.

*Code and data availability.* The authors thank the AmeriFlux team for making the eddy-covariance flux data available for this study. The FLUXNET2015 datasets used in this study have been downloaded from the FLUXNET community data portal (http://fluxnet.fluxdata.org/data/fluxnet2015-dataset/). The version of SCOPE model used in this study can be obtained from https://github.com/Christiaanvandertol/SCOPE.



## Appendix A: Modeling Photosynthesis in SCOPE

The biochemical module is at the center of energy balance computations within SCOPE. This module computes the net assimilation (photosynthesis), stomatal conductance and the chlorophyll fluorescence of a leaf. This module is thus extremely important because the coupled photosynthesis and stomatal conductance regulates the latent heat flux which in turn affects the

net energy balance and the leaf temperature, which in turn again affects the leaf photosynthesis and subsequently the energy balance. SCOPE computes the leaf temperature and the overall energy balance iteratively such that they there is closure in energy balance. As such the leaf temperature and its regulation of photosynthesis forms an extremely important component of the overall energy balance of the canopy. In this study we have made changes to the biochemical module of SCOPE to make it consistent with the widely used CLM4.5. We have adapted photosynthetic model together with coupled temperature

dependence of the photosynthetic parameters according to the implementation in CLM4.5. This includes both the temperature dependence functions and the high temperature inhibition of the parameters. The model includes exclusive pathways both for the $C_3$ and $C_4$ plant species and is represented as follows:

The net photosynthesis (assimilation) after accounting for respiration ($R_d$) is given as:

$$A_n = \min(A_c, A_j, A_p) - R_d \tag{A1}$$

Further, the rate limiting steps are represented as follows:

The RuBP carboxylase (Rubisco) limited rate of carboxylation $A_c$ is given by:

$$A_c = \begin{cases} \frac{V_{cmax}(C_i - \Gamma^*)}{C_i + K_c(1 + O_i/K_o)}, & \text{for } C_3 \text{ species.} \\ V_{cmax}, & \text{for } C_4 \text{ species.} \end{cases} \tag{A2}$$

The light-limited rate of carboxylation (governed by the capacity to regenerate RuBP) $A_j$ is given by:

$$A_j = \begin{cases} \frac{J(C_i - \Gamma^*)}{4(C_i + 2\Gamma^*)}, & \text{for } C_3 \text{ species.} \\ \alpha(4.6\phi), & \text{for } C_4 \text{ species.} \end{cases} \tag{A3}$$

Finally the product limited carboxylation rate for $C_3$ plants and the PEP-carboxylase-limited rate of carboxylation for the $C_4$ plants $A_p$ is given by:

$$A_p = \begin{cases} 3T_p, & \text{for } C_3 \text{ species.} \\ k_p C_i, & \text{for } C_4 \text{ species.} \end{cases} \tag{A4}$$

For the above equations A2, A3, A4, we have the assimilation rates $A_{c,j,p}$ in the units of $\mu$mols m$^{-2}$ s$^{-1}$, $C_i$ is the internal $CO_2$ concentration of the leaf (units of ppm) and $V_{cmax}$ is the maximum rate of carboxylation. For the $C_3$ species, $K_c$ and $K_o$





are the Michelis-Menten constants for $CO_2$ and $O_2$ respectively (units of $\mu$mols m$^{-2}$ s$^{-1}$), $\Gamma^*$ is the $CO_2$ compensation point (units are ppm), $J$ is the potential electron transport rate (units of $\mu$mols m$^{-2}$ s$^{-1}$) and $T_p$ is the triose phosphate utilization rate. For the C$_4$ plants, $\phi$ is the absorbed PAR in the units of Wm$^{-2}$ and the factor 4.6 converts it to PPFD in units of $\mu$mol m$^{-2}$ s$^{-1}$ (for SCOPE biochem module the PAR is already in PPFD units), $\alpha$ is the quantum efficiency (0.05 mol $CO_2$ mol$^{-1}$ photon), and $k_p$ is the initial slope of the C$_4$ $CO_2$ response curve.

For the C$_3$ plants, the potential electron transport rate $J$ depends on the PAR absorbed by a leaf, which is obtained as the smaller root of the two roots of the equation:

$$\Theta_{PSII}J^2 - (I_{PSII} + Jmax)J + I_{PSII}J_{max} = 0 \tag{A5}$$

Where, $J_{max}$ is the maximum electron transport rate ($\mu$mols m$^{-2}$ s$^{-1}$), $I_{PSII}$ is the light used in photosystem II ($\mu$mols m$^{-2}$ s$^{-1}$) which is given by eqn. A6 and $\Theta_{PSII}$ is a curvature parameter.

$$I_{PSII} = 0.5\Phi_{PSII}(4.6\phi) \tag{A6}$$

The term $\Phi_{PSII}$ in eqn. A6 is the quantum yield of photosystem II and 0.5 represents half electron transfer to each of the photosystems I and II. The overall gross photosynthesis rate is computed as a co-limitation (Collatz et al., 1991a, 1992) and is computed as the smaller root of the equations:

$$\Theta_{cj}A_i^2 - (A_c + A_j)A_i + A_cA_j = 0$$
$$\Theta_{ip}A^2 - (A_i + A_p)A + A_iA_p = 0 \tag{A7}$$

The parameters $\Theta_{cj}$ and $\Theta_{ip}$ control the smoothness of the light response curve between light limited and enzyme/product limiting rates. The values of the different parameters at optimum temperature (mostly as a function of $V_{cmax25}$ here the optimum temperature, is assumed as 25$^\circ$C) used in the photosynthesis model are presented in Table A1

## Appendix B: Temperature Dependence of Photosynthetic Parameters

The photosynthesis model parameters for both the C$_3$ and C$_4$ pathways described in the previous section and shown in Table A1 have temperature dependent variations and need to be adjusted for specific leaf temperature before implementing them in the photosynthesis model. The temperature dependence of photosynthetic parameters for the C$_3$ species can be broadly decomposed into two parts (i) the temperature response and (ii) the high temperature inhibition. The functional form of these are as follows:





| Parameter | $C_3$ | $C_4$ |
|---|---|---|
| $R_d^{opt}$ | $0.015 V_{cmax}^{opt}$ | $0.025 V_{cmax25}$ |
| $J_{max}^{opt}$ | $1.97 V_{cmax}^{opt}$ | - |
| $K_c^{opt}$ | $404.9$ | - |
| $K_o^{opt}$ | $278.4$ | - |
| $\Gamma_{opt}^*$ | $\frac{0.5 O V_o K_c}{V_c K_c}$ | - |
| $T_p^{opt}$ | $0.1182 V_{cmax}^{opt}$ | - |
| $\Theta_{PSII}$ | $0.7$ | - |
| $\Phi_{PSII}$ | $0.85$ | - |
| $k_p^{opt}$ | - | $20000 V_{cmax}^{opt}$ |
| $\Theta_{cj}$ | $0.98$ | $0.80$ |
| $\Theta_{ip}$ | $0.95$ | $0.95$ |

**Table A1.** Functional forms of photosynthesis equation parameters

$$f(T_v) = exp\left[\frac{H_a}{RT_0}\left(1 - \frac{T_0}{T_v}\right)\right]$$

$$f_H(T_v) = \frac{1 + exp\left(\frac{S_v T_0 - H_d}{RT_0}\right)}{1 + exp\left(\frac{S_v T_v - H_d}{RT_v}\right)} \tag{B1}$$

Where, $H_a$ is the activation energy, $H_d$ is the deactivation energy, $S_v$ is the entropy term, $T_o$ is the optimum temperature and $T_v$ is the leaf temperature. The functional relationship of the different photosynthetic parameters in the $C_3$ pathway are as follows:

$$V_{cmax} = V_{cmax}^{opt} f(T_v) f_H(T_v)$$

$$J_{max} = J_{max}^{opt} f(T_v) f_H(T_v)$$

$$T_p = T_p^{opt} f(T_v) f_H(T_v)$$

$$R_d = R_d^{opt} f(T_v) f_H(T_v)$$

$$K_c = K_c^{opt} f(T_v)$$

$$K_o = K_o^{opt} f(T_v)$$

$$\Gamma^* = \Gamma_{opt}^* f(T_v) \tag{B2}$$





The temperature dependence of photosynthetic parameters for the $C_4$ species are given by the following relationships:

$$V_{cmax} = V_{cmax}^{opt} \left[ \frac{f(Q_{10})}{f_U(T_v) f_L(T_v)} \right]$$

$$f(Q_{10}) = Q_{10}^{(T_v - T_0)/10}$$

$$f_U(T_v) = 1 + exp(s_1(T_v - s_2))$$

$$f_L(T_v) = 1 + exp(s_3(s_4 - T_v)) \tag{B3}$$

$$R_d = R_d^{opt} \left[ \frac{f(Q_{10})}{f_U(T_v)} \right] \tag{B4}$$

$$k_p = k_p^{opt} f(Q_{10}) \tag{B5}$$

5   The $Q_{10}$ temperature coefficient is a measure of the rate of change of a biological or chemical system as a consequence of increasing the temperature by 10°C. The values of the temperature dependence functional parameters for both $C_3$ and $C_4$ species used in the present study are provided in tables B1 and B2 respectively.

| Parameter | $H_a$ (J mol$^{-1}$) | $H_d$ (J mol$^{-1}$) | $S_v$ (J mol$^{-1}$ K$^{-1}$) | $T_0$ (K) |
|---|---|---|---|---|
| $V_{cmax}$ | 65330 | 14920 | 485 | 298 |
| $J_{max}$ | 43540 | 152040 | 495 | 298 |
| $T_p$ | 65330 | 14920 | 485 | 298 |
| $R_d$ | 46390 | 150650 | 490 | 298 |
| $K_c$ | 79430 | - | - | 298 |
| $K_o$ | 36380 | - | - | 298 |
| $\Gamma^*$ | 37830 | - | - | 298 |

**Table B1.** $C_3$ Temperature dependence functional parameters

| Parameter | $Q_{10}$ (-) | $s_1$ (K$^{-1}$) | $s_2$ (K) | $s_3$ (K$^{-1}$) | $s_4$ (K) |
|---|---|---|---|---|---|
| $V_{cmax}$ | 2 | 0.3 | 313.15 | 0.2 | 288.15 |
| $R_d$ | 2 | 1.3 | 328.15 | - | - |
| $k_p$ | 2 | - | - | - | - |

**Table B2.** $C_4$ Temperature dependence functional parameters

The temperature dependence parameters (activation, deactivation and entropy) is variable between different plant species (Leuning, 2002) as such its formulation in the newer implementation of the SCOPE model allows us to use appropriate values

10   depending on the ecosystem we study.



## Appendix C: Derivation of Iterative Retrieval Algorithm

For deriving the maximum probability state $X$ ($\hat{X}$) we equate the derivative of the equation 8 to zero to obtain:

$$\nabla_X\{-2\ln P(X|Y)\} = -[\nabla_X F(X)]^T S_\epsilon^{-1}[Y - F(X)] + S_a^{-1}[X - X_a] = 0 \tag{C1}$$

It can be noted here that the gradient $\nabla_X$ of the above vector valued function is a matrix valued function and the Jacobian

matrix is represented as: $K(X) = \nabla_X F(X)$ and which results in the following implicit equation for $\hat{X}$:

$$-K^T(X)S_\epsilon^{-1}[Y - F(X)] + S_a^{-1}[X - X_a] = 0 \tag{C2}$$

We have to now use any general root finding method for finding the solutions of equation C2. If the problem is not too non-linear we can use the Newton and Gauss-Newton iterative methods (Hartley, 1961). In general for any vector equation $G(X) = 0$, we can write the Newton iteration as follows:

$$X_{i+1} = X_i - [\nabla_X G(X_i)]^{-1} G(X_i) \tag{C3}$$

For our problem we can assume the derivative of the cost-function $G(X)$ to be the LHS of equation C1, therefore the gradient of $G(X)$ ($\nabla G$) also known as the Hessian is given by:

$$\nabla_X G(X) = S_a^{-1} + K^T S_\epsilon^{-1} K - [\nabla_X K^T] S_\epsilon^{-1}[Y - F(X)] \tag{C4}$$

The Hessian in equation C4 involves the Jacobian $K$ and both the first and second derivatives of the forward model. The

second derivative is complicated because it is a vector whose elements are matrices and further this term is post multiplied by the factor $S_\epsilon^{-1}[Y - F(X)]$. The third term in the RHS of equation C4 is thus computationally expensive and further for moderately linear problems this term is small, as such this term can be ignored (also called small-residual problems in numerical methods). When we ignore this term, we get the Gauss-Newton iteration scheme by substituting equations C2 and C4 in equation C3:

$$X_{i+1} = X_i + (S_a^{-1} + K_i^T S_\epsilon^{-1} K_i)^{-1}[K_i^T S_\epsilon^{-1}[Y - F(X_i)] - S_a^{-1}[X_i - X_a]] \tag{C5}$$

where, $K_i = K(X_i)$, we can substitute $F(X)$ from equation 7 in equation C2 to get:

$$-K^T(\hat{X})S_\epsilon^{-1}[Y - F(X)_{X=X_l} + \nabla_X F(X)_{X=X_l}(X - X_l)] + S_a^{-1}[\hat{X} - X_a] = 0 \tag{C6}$$




Again, representing $K_l = \nabla_X F(X)_{X=X_l}$, $F_l = F(X)_{X=X_l}$ we can further simplify and rearrange equation C6 as:

$$S_a^{-1}[\hat{X} - X_a] + K_l^T S_\epsilon K_l(\hat{X} - X_a) = K_l S_\epsilon^{-1}[Y - F_l + K_l(X_l - X_a)]$$
$$\hat{X} = X_a + (S_a^{-1} + K_l^T S_\epsilon^{-1} K_l)^{-1} K_l S_\epsilon^{-1}[Y - F_l + K_l(X_l - X_a)] \tag{C7}$$

In the above equations, if we change the interpretation of the subscript $l$ from 'linearization' to 'iteration counter', we obtain the following equation:

$$X_{i+1} = X_a + (S_a^{-1} + K_i^T S_\epsilon^{-1} K_i)^{-1} K_i S_\epsilon^{-1}[Y - F(X_i) + K_i(X_i - X_a)] \tag{C8}$$

If we express $X_{i+1}$ as a departure from $X_i$ rather than $X_a$ we obtain the same expression for the iteration steps as equation C5 or 9.

*Competing interests.* The authors declare no competing interests.

*Acknowledgements.* The research was carried out, in part, at the Jet Propulsion Laboratory, California Institute of Technology, under a contract with the National Aeronautics and Space Administration. California Institute of Technology. Government sponsorship is acknowledged. ©2018. All rights reserved.



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
