# Peer review of "Optimal Inverse Estimation of Ecosystem Parameters from Observations of Carbon and Energy Fluxes"

_Biogeosciences, 2018_

## Referee Comment (RC1) · Anonymous Referee #1 · 18 Aug 2018

MAIN COMMENTS

The article presents a methodology to estimate parameters of a land-surface carbon and energy balance model (SCOPE) through a Bayesian non-linear inversion framework. First, the biochemical model of photosynthesis is modified to resemble the model implemented in the land-surface model CLM4.5. Then, the values of three important parameters: Vc,max, BBslope (the slope of the empirical function relating net assimilation to stomatal conductance) and LAI are computed for one growing season for three locations characterized by different vegetation types, a C4 crop, a C3 deciduous forest and a C3 evergreen forest. The methodology leads to estimate the seasonality

of the analyzed parameters (Fig. 9, 14 and 18) and improve model performance in reproducing carbon and energy fluxes (Fig. 12 and 16).

The article is generally well written, although with a large number of imprecisions (see detailed comments). From my evaluation the technical aspects of the inversion algorithm to estimate the parameters are rigorously implemented. However I have a number of concerns which are detailed in the major comments below.

(i) The algorithm of the Bayesian inversion framework is designed to provide robust numerical results and optimize the model parameters to reduce the mismatch between observations and simulations. However, the same results are likely less meaningful from a plant physiological point of view. This is important given the premises of the authors in giving a physiological interpretation of the parameters (Page 2, LL 7-9). When I see (a) Vc,max changing from less than 10 to 80 throughout the growing season (Fig. 9 and 14), (b) LAI decreasing from 5-6 to less than 4 in the second part of the growing season for a crop, which is expected, at the very least, to maintain the same LAI until harvest (Fig. 9), or (c) a threefold variability in BBslope, I tend to think, there is a considerable adjustment of model structural issues rather than an estimation of meaningful ecosystem parameters. This is hinted by the authors when they say that they do not include evaporation from interception or have a simplified soil evaporation (Page 20, LL 2-4). Surely, Vc,max or photosynthetic capacity have been observed to change seasonally even considerably in a single tree (e.g., Wilson et al 2001; Misson et al 2006; Bauerle et al 2012; Wu et al 2017) but not in a range that cover almost the entire variability of Vc,max globally (Kattge et al 2009). What would be the physiological explanation for such a variation in Vc,max? The seasonality of leaf nitrogen content is surely much less pronounced. I think what we see in the seasonality of parameters is largely a compensation of model structural inadequacy (e.g., the jumps in correspondence to major rainfall events) and only partially a real seasonality. Better constraints should be places on the potential range of the parameters. Furthermore, some of these "parameter" as the seasonality of LAI could be tested against ground

or remote sensing observations to support or disprove the values obtained in the optimization. It is also not surprising that model results are improved (even though it is not clear how much, since R2 results with the non-optimized model are not reported) since now the parameter space is significantly larger, being parameters allowed to vary seasonally.

(ii) A major issue is also the use of a single year for the three locations. In this way, the robustness of such estimates across different years remain untested. In my view, an important test, would be to use the constant and seasonal parameters over few other years and see if the prediction are better in the case of seasonally variable parameters. Saying that they are better when they are optimized is completely expected and trivial. Finally, I am glad the author consider uncertainty in the flux-tower data ($S\varepsilon$), which is a very important aspect given the considerable uncertainties in flux-tower observations, however, how they do (Page 16, Line 2-3) is not very clear nor justified.

(iii) I also have doubts about the overall novelty of the study. Several other studies have been published using Bayesian approaches to parameter estimation with inverse methods in ecosystem or land-surface models (e.g., Mackay et al 2012, Xu et al 2006; Wu et al 2009; Wolf et al 2006). For instance, the authors do a remarkable job in highlighting the subtle differences of their work with respect to Wolf et al 2006 (Page 3, LL 15-21). Surely, in this article, the model is different, the optimized parameters are different and the inversion methodology has some peculiarity but overall the idea and scope is not much dissimilar from the ones of previous studies. If the novelty is on technical aspects of the methodology, then "Biogeosciences" may not be the most appropriate venue.

(iv) Finally, some of the presented material is redundant. There is an entire manuscript part in comparing new and old implementations of the biochemical model of photosynthesis (Section 2.3 Fig. 2, 3, 4 and 5), which is just relevant for the SCOPE model users, since it is quite obvious that if one substitutes temperature functions in the biochemical photosynthesis module, he/she might obtain different results. These type of

analyses are carried out by any model developer all the time, without the necessity to write 5 pages of peer-reviewed paper about it. Figure 11 is also very technical and I think it would be more appropriate for an appendix or supplementary material than for a main text. The results described for the three sites are also separated and many explanatory sentences are repeated. Their presentation can be largely streamlined, combining Figure 8, 13 and 17, Figure 9, 14 and 18, and Figure 10, 15 and 19 and removing the repetitive parts, highlighting differences among the case studies, rather than iterating the overall result presentation.

DETAILED COMMENTS

Page 2 LL 5. See also Wramneby et al 2008; Pappas et al 2013.

Page 2 LL 18-19. Which model do you refer to? LSM, Ecosystem/Vegetation models or the biochemical models of photosynthesis? The first they also require shortwave radiation and precipitation as input and information about soil depth and properties, at the very least.

Page 2. LL 26. Rate-limiting, strictly speaking, refers only to Jmax not Rd.

Page 2. LL 33-34. LAI in most of Terrestrial Biosphere Models is a prognostic variable not a parameter. As you correctly wrote in Page 3, LL 13-14. I think, this needs to be stated here.

Page 3. LL 2. LAI can be also estimated from destructive observations.

Page 3. LL 12 and Page 4 LL 12. SCOPE can model "spectrally resolved" radiation, but it remains unclear throughout the manuscript how many wavebands and which ones are considered?

Page 3. LL 22 See also Mackay et al 2012.

Page 3. LL 25. I would write "often the associated computational costs..."

Page 3. LL 26-31. While I completely agree that modeling SIF and comparing SIF with

observations is very important, since SIF is not explicitly used in this manuscript, such a long paragraph in the introduction, deviates from the main focus of the article.

Page 4. LL 26-28. So, in the end, how many prognostic temperatures do you resolve in the system? E.g., 2 temperatures for each layer for how many layers?

Page 4. LL 28. Since you brought this up and this is not an easy problem to solve. Are iterations repeated until convergence? Which tolerance is used for convergence? If not, how many iterations are used?

Page 5. LL 10-16. I found this part explaining differences with CLM4.5 at least awkward. As a reader I want to know what you do now, not what was different from CLM4.5 in the previous model version.

Page 5. LL 20-21. Please move to an earlier part of the section the explanation of what CLM is.

Page 6. LL 1. There is no mention of the parameter "intrinsic quantum use efficiency" or "quantum yield of photosystem II" depending how it is expressed. While it is considered constant by most biochemical models, this could also exhibit some variability (e.g., Skillman 2008) and could have been integrated in the optimization framework.

Page 6. LL 4-5. As before, please delete the second part of the sentence "in the earlier SCOPE version it was implemented as potential ETR x CO2 per electron", this is not relevant here.

Page 6. LL 15-20. Cs is not defined, but it must be the CO2 concentration on the leaf-surface, otherwise a leaf boundary layer resistance would be necessary in Eq. (1) and (2).

Page 6. LL 26. Iterative methods to solve the A – Ci – gs system were already included in ecohydrological models, (e.g., Ivanov et al 2008 and other land-surface models before that).

Page 7. LL 1. This is repetitive, it has been already stated a few times.

Page 7, LL 6-7 and Figure 1, Caption. It is not clear what $\pm\sigma$ variability means. It is the variability in the parameters of the temperature functions, how $\sigma$ is defined? Is the standard deviation of which parameter? What data from Leuning 2002 are considered/used for the plots? Are the temperature parameters rather than data taken from Leuning, 2002?

Page 7. Line 16. The two sites are not defined yet, their description is arriving much later in the manuscript, while it should be made upfront.

Page 8. Line 5, Figure 5 and Page 11 Line 4-6. Which version has been calibrated to the data? It is not surprising that the newer implementation is closer to observations, if Vc,max or other parameters have been optimized for the newer version. For instance, in Figure 5b you could likely adjust the value of Vc,max to obtain an opposite behavior, where the old model is unbiased and the new one is.

Page 8. Line 9-10 and Figure 3 and 4 caption. I do not think you are showing any result for the Missouri Ozark site or Nebraska-Mead-1, you just use the meteorological forcing and C3 vegetation type derived for these sites to run the two version of the biochemical model and make a comparison. Strictly speaking you could have done a test varying temperature, VPD, and PAR for C3 and C4 vegetation without referring to any specific site. However, I do not find this part generally insightful for a journal article.

Page 13. Line 4. The Jacobian depends on the linearization point, X_l, which is somehow approximated for any optimization window. This is discussed later but maybe it should be stated here already.

Page 13. Line 14-15. It is fine to have a more in-depth treatment in the appendix, but at least you need to define the terms, Ki (the Jaobian matrix) is never defined at this stage of the article.

Page 14, Figure 6, and also Page 22, LL 5-6. Some of the derivative values are

a bit surprising. Why LE is decreasing with increasing LAI? Is due to self-shading effects? This is generally counterintuitive. Even more difficult to understand is why LE decreases with increasing Vc,max. Higher photosynthesis should lead to higher stomatal conductance and thus higher LE, especially in a model like SCOPE. These unexpected behaviors need justification.

Page 15. Figure 7 caption. You need to specify what is "m" and what is intended here for $\Delta X$. Plus, it should be better stated what the measurement vector refers to, since the derivative must be computed with model outputs.

Page 16. Line 2-3. The observational error matrix $S_\varepsilon$ is quite important given the general uncertainty of flux-tower observations. However, from such a brief description "using noise standard deviation as 10% of observations" is not clear how this is computed. Does it mean that you assume an uncertainty of 10%? This is likely quite a low number in the context of flux-tower observations (e.g., Leuning et al 2012).

Page 16. Line 5. Which iteration step? The current one? This is relevant since K depends on where it is computed.

Page 16. Line 6-7. I am not sure I can see very well the concatenation between observations Y and modeled values F(X) in Figure 7. I just see $\Delta Y$.

Page 16. Line 20-22. I agree with the authors that 3-days sounds as a reasonable length for separating the time scales of parameter variability and meteorological forcing. But, what does it happen if you modify the time window and instead of 3 days you select 7 days or 15 days? Do you get similar seasonality of parameters? This is a test, which could be relevant in the scope of this article.

Page 16. Line 24. I do not find where this is mentioned in Section 5.

Page 17. Equation (17). The subscript "jj" is undefined.

Page 18. Figure 8. Did you see that incoming shortwave radiation is reconstructed and actually almost equal for the first 60 days? Afterwards, fortunately, you do not use this

period because it is not in the growing season, but such type of artifacts, which are frequent in Fluxnet data, could jeopardize your procedure. This may need a mention.

Page 18. Line 8. Why did you select a single year? I think it would be rather important to see how seasonality of parameters is retrieved in different years.

Page 19. Line 8-9, Please provide references for such changes in BBslope, LAI and Vc,max if they are found to be reasonable, which I do not think it is the case for BBslope and Vc,max

Page 20. Line 12 and captions of Figure 10 and also Figure 15 and Figure 19. If it is a "correlation coefficient", why the symbol r2 is used, this is typically reserved for the "coefficient of determination", and not for the correlation coefficient, which is typically indicated by r.

Page 21. Figure 11. The x-label with the indexes is not defined in the figure caption but only in the main text.

Page 22. LL 7 and LL 10 and Figure 12 caption. How much is it this "net improvement"? The value of R2 for the previous parameter set is not reported, values are only reported for the optimized model.

Page 23. LL 12. It is quite well-known that the growing season is longer for a deciduous forest than for a crop. I would suggest eliminating "we find".

Page 24, LL 13 and Page 29, LL 6-7. What could justify a threefold change of BBslope in a single growing season? This is theoretically an intrinsic property of the stomatal regulation, I can see how can change during leaf development (very first part of the growing season) or if water stress occur, but I do not see what can justify such a large variability throughout the entire season.

Page 25. LL 13-16. The value of R2 for the previous parameter set is not reported, values are only reported for the optimized model. How much the results were improved remains unverifiable, although clear from Figure 16.

Page 27. LL 2-6. Please consider that at Niwot Ridge snow-cover is affecting energy exchange and potentially GPP for a large fraction of the year. It is true that you focus only in the snow free-period, but this confounding element needs to be stated somewhere.

Page 27. LL 10. The constraint on LAI from observation is a very good addition to the modeling exercise, I would have liked to see this type of constraints placed also for other sites, or parameters, whenever available.

Page 29. LL 16 and Page 30 LL 4-5. I am sorry, but I am thinking you mostly retrieve "model parameters" and not "ecosystem parameters", and I also tend to think you are "overfitting" the SCOPE model rather than constraining it. The advantage of using seasonality of parameters for predictive simulations (e.g., in other years or other conditions) remain to be tested.

Page 29. LL 22. Yes, LAI and $V_{c,max}$ have seasonal variability but for $V_{c,max}$ unlikely in the order of magnitude that is presented here.

Page 29. LL 27-30. While interesting the discussion on SIF is out of place, since SIF is not used or treated in this article.

Page 30. LL 16. Optimality approaches (e.g., Medlyn et al 2011, Katul et al 2010) do not typically comprise soil-moisture dependences, which therefore need to be included as additional parameterizations.

Page 30. LL 18. Rather than "better solution", I would say it can provide "optimized model parameters."

Page 30. LL 19-25. All this paragraph is emphasizing the technical aspects of the "optimal inverse estimation of the parameters", I am wondering if this is the most effective way of concluding the manuscript.

Page 31. LL 7-8. This sentence is quite repetitive.

Page 34. LL 9. I would suggest to write "will allow us".

References

Bauerle, W. L., R. Oren, D. A. Way, S. S. Qian, P. C. Stoy, P. E. Thornton, J. D. Bowden, F. M. Hoffman, and R. F. Reynolds (2012), Photoperiodic regulation of the seasonal pattern of photosynthetic capacity and the implications for carbon cycling, Proc. Natl Acad. Sci. USA, 109 (22), 86128617.

Ivanov, V.Y., Bras, R.L., Vivoni, E.R., 2008. Vegetation-hydrology dynamics in complex terrain of semiarid areas. I: a mechanistic approach to modeling dynamic feedbacks. Water Resour. Res. 44, W03429. http://dx.doi.org/10.1029/2006WR005588.

Kattge J, Knorr W, Raddatz T, Wirth C. 2009. Quantifying photosynthetic capacity and its relationship to leaf nitrogen content for global-scale terrestrial biosphere models. Global Change Biology 15: 976–991.

Katul G, Manzoni S, Palmroth S, Oren R. A stomatal optimization theory to describe the effects of atmospheric CO2 on leaf photosynthesis and transpiration. Ann Bot 2010, 105:431–442.

Leuning R, van Gorsel E, Massman WJ, Isaac PR.2012 Reflections on the surface energy imbalance problem. Agr Forest Meteorol, 156:65–74. doi:10.1016/j.agrformet.2011.12.002

Mackay DS et al (2012). Bayesian analysis of canopy transpiration models: a test of posterior parameter means against measurements. Journal of Hydrology 432–433 75–83

Medlyn BE, Duursma RA, Eamus D, Ellsworth DS, Prentice IC, Barto CVM, Crous KY, De Angelis P, Freeman MC, Wingate L. Reconciling the optimal and empirical approaches to modelling stomatal conductance. Glob Change Biol 2011, 17:2134–2144.

Misson, L., K. P. Tu, R. A. Boniello, and A. H. Goldstein (2006), Seasonality of photosynthetic parameters in a multi‐specific and vertically complex forest ecosystem in the Sierra Nevada of California, Tree Physiol., 26, 729–741

Pappas, C., et al (2013), Sensitivity analysis of a process-based ecosystem model: Pinpointing parameterization and structural issues, J.Geophys. Res. Biogeosci., 118, doi:10.1002/jgrg.20035.

Skillman, J. B. (2008), Quantum yield variation across the three pathways of photosynthesis: not yet out of the dark, Journal of Experimental Botany, 59 (7), 1647-1661.

Wilson, K.B., D.D. Baldocchi and P.J. Hanson. 2001. Leaf age affects the seasonal pattern of photosynthetic capacity and net ecosystem exchange of carbon in a deciduous forest. Plant Cell Environ. 24:571–583

Wolf, A., Akshalov, K., Saliendra, N., Johnson, D. A., and Laca, E. A. 2006 Inverse estimation of Vc max, leaf area index, and the Ball-Berry parameter from carbon and energy fluxes, Journal of Geophysical Research, 111, 1–18,

Wramneby, A., B. Smith, S. Zaehle, and M. T. Sykes (2008), Parameteruncertainties in the modelling of vegetation dynamics – Effects on tree community structure and ecosystem functioning in European forest biomes, Ecol. Model., 216(3-4), 277–290, doi:10.1016 /j.ecolmodel.2008 04.013.

Wu XW, Luo YQ, Weng ES, White L, Ma Y, Zhou XH (2009) Conditional inversion to estimate parameters from eddy-flux observations. J Plant Ecol 2:55–68

Wu, J., Serbin, S. P., Xu, X., Albert, L. P., Chen, M., Meng, R., et al. (2017). The phenology of leaf quality and its within-canopy variation are essential for accurate modeling of photosynthesis in tropical evergreen forests. Global Change Biology, 23, 4814–4827

Xu T, White L, Hui D, Luo Y. 2006. Probabilistic inversion of a terrestrial ecosystem model: analysis of uncertainty in parameter estimation and model prediction. Glob. Biogeochem. Cycles 20:GB2007

---

## Author Comment (AC1) · 25 Aug 2018

We thank the referee for the careful, detailed and knowledgeable review of the manuscript. While the review is critical of some of our analysis, we are grateful as the comments will improve the quality of our submission. In the spirit of the open journal, we wanted to write a high-level response before the discussion phase closes (with a point by point response to all individual concerns once we gathered all available reviews).

There is a general concern about the Vcmax variability (and some other parameters). We would like to note that some of the papers cited by the reviewer and others (e.g. Wilson et al 2001) show considerable Vcmax variability, even in systems with almost constant N content. These changes are mostly attributed to substantial in-season changes in the fraction of total N allocated to Rubisco as well as changes in LMA. In addition, most models have no other method of imposing environmental stress than reducing Vcmax by a stress factor [0,1]. The effect of reductions in Vcmax are a reduction in A, which also suppresses transpiration (and stress might also change the Ball Berry slope). We should probably make it clearer that we are fitting an "effective Vcmax" parameter, which factors in effects from true changes in Rubisco content as well as the impact of stress.

As for novelty: We admit that we haven't yet used the full potential of SCOPE in the current manuscript, which is our main motivation to use SCOPE as we eventually want to make use of remote sensing data as well (the current setup could have been performed with simpler models). We will add a better description of the potential of using modeled reflectance from SCOPE, in particular for constraining LAI. It will be outlined with a few inversion examples as well, which should alleviate the concern regarding novelty. We will also streamline the narrative and potentially move some sections in the supplementary. Many thanks again for the detailed review.

---

## Referee Comment (RC2) · Anonymous Referee #2 · 9 Sep 2018

The present study attempts to develop an optimal inversion framework to use SCOPE for estimating Vcmax, m, and LAI by against measurements of carbon and energy flux from EC towers. They demonstrated the applicability of their approach in terms of capturing the seasonal variability of these key ecosystem parameters.

The current work may provide additional information on estimating key ecosystem parameters from field data. Compared to the literatures, however, the novelty of the current study is not clear. There are so many papers which estimates the key ecosystem parameters from models and EC flux tower data (e.g., Mackay et al 2012, Xu et al 2006; Wu et al 2009; Wolf et al 2006; Wang et al., 2010). What is the main novelty

for this work? If it is the technical approach of an optimal inversion framework, then it may go some other technical journal. Even for the inversion framework, I didn't see too much improvements compared to previous work.

My general impression is that this work is a rather technical description on the inversion framework of using SCOPE. I understood the rather detailed information by the authors, but the manuscript is really too long and some parts are too lengthy. I think some parts could be simplified.
* * *

---

## Referee Comment (RC3) · P. Rayner (Referee) · 11 Sep 2018

This paper demonstrates a method for assimilating site-level flux observations into a terrestrial biosphere model. Its novelty lies in breaking the assimilation into short windows to capture high-frequency variations in the parameters it estimates. given the variety of journals within the Copernicus family, I wonder whether this article is better suited to GMD than BG (see comments below) but this is mainly a question for the editor. the paper is also clearly written, verging on the tutorial at times.

I have one significant concern with the paper and one general request for more analysis. My concern is the analysis of the results. This is quite thin. The only commentary

I can see on the results in the discussion section is: "There is strong evidence from measurements that under normal conditions LAI and photosynthetic parameters have seasonal variability [Wang et al., 2008; Wilson and Bal- docchi, 2000; Wilson et al., 2000] which correlate with observations of energy fluxes. Our model inversion results are in alignment and agree well with these observations." this seems quite a poor scientific return from a difficult and well-executed piece of work. I would recommend particularly using the posterior simulation to look at some other observables. Do you do a better job matching the high SIF values over the corn site? If so, why, e.g.which parameter, Vcmax or LAI is mainly responsible? What temporal resolution of the parameters is necessary to capture the important variations? I suspect these questions only scratch the surface. I stress that this is potentially a good paper. What it does it does well but I believe it needs more scientific content before publication. If the authors wish to maintain it near its present form I believe it is better suited as a demonstration of a new methodology and hence to GMD.

My request is to delve a little deeper into why the system works better at some places than others. I note there seems less analysis of the Niwot Ridge results which were, in general, also less successful (lower correlation for example). Remember that a less successful assimilation is *not* a failure but rather a useful probe into model performance. It says definitively "we have a problem here and it isn't the choice of parameters". This is even clearer in this case where the parameters are allowed to vary in time.

Minor comments P14 In fact the Jacobian doesn't quite show the problem is non-linear, it could be that all the variation is a result of different forcing.

P16L3 The choice of observational error is quite important in DA, hoefuly this is checked later.

P16L10 I doubt the size of observational vector has much impact on computational efficiency, can you comment why it would?

P16L20 The choice of time resolution is also important and yours seems very short. This is likely to lead to parameters which can vary fairly rapidly in time but which are also quite uncertain as they are constrained by fewer observations. Hopefully you can comment on whether parameters change significantly, i.e outside their uncertainty limits.

P16 Eq. 12, this should have a term from the prior included I think. Unless there's no prior.

P19L7 "reasonable and realistic" is a little vague, perhaps some references would help

P20L10 be careful about describing correlations as describing how parameters move since these are uncertainty not signal correlations. the sentence above makes it clear you understand this difference but many of your readers will be less clear.

P20L14 but here you do confuse signal and error, this correlation does NOT indicate they are changing in sync

P20 in general you seem to be quoting r^2 but claim this can be negative. You probably mean r.

P21 I'm not sure that the figures showing your algorithm works are necessary, especially in a journal like biogeosciences where you should focus more on the science and less on the algorithm.

P22L3 as noted earlier the diurnality is not a measure of nonlinearity

P22L10 don't quote improved correlation as a measure of fit, you could have a great correlation and terrible performance if, for example, diurnal variations had great phase and terrible amplitudes. rms is a better though not perfect statistic.

P25 See earlier comments on signal and error correlation.

P25 can you explain further why a strong negative correlation means you need to optimise both, the step from "you can't see them separately" to "you must do both of

them" isn't so clear to me

P25 I hope you go on to compare the performance at the two sites, one of them seems much harder than the other.

P27 I'm betting you originally tried to fit LAI at NWR and couldn't. That's not a failure, it's interesting information so is probably worth discussing. You're only fitting in 3 day windows so neither site really knows about the evolution of LAI from one window to the next so why does one work well and the other not, provided I'm guessing correctly.

P29L13 This site analysis doesn't seem as well developed as the others, e.g. quality of fit etc.

P29L30 do you mean changes in the temperature dependencies or more simply that there *is* a temperature dependence?

P30L20 In what sense is the approach "stepwise"? This term was previously used by Bacour et al. (2015), doi:10.1002/2015JG002966) to describe optimising for one observable then using its posterior parameters as priors for the next observable. They would describe your method as "all at once", what do *you* mean by stepwise?

This paper demonstrates a method for assimilating site-level flux observations into a terrestrial biosphere model. Its novelty lies in breaking the assimilation into short windows to capture high-frequency variations in the parameters it estimates. given the variety of journals within the Copernicus family, I wonder whether this article is better suited to GMD than BG (see comments below) but this is mainly a question for the editor. the paper is also clearly written, verging on the tutorial at times.

I have one significant concern with the paper and one general request for more analysis. My concern is the analysis of the results. This is quite thin. The only commentary I can see on the results in the discussion section is: "There is strong evidence from measurements that under normal conditions LAI and photosynthetic parameters have seasonal variability [Wang et al., 2008; Wilson and Bal- docchi, 2000; Wilson et al.,

2000] which correlate with observations of energy fluxes. Our model inversion results are in alignment and agree well with these observations." this seems quite a poor scientific return from a difficult and well-executed piece of work. I would recommend particularly using the posterior simulation to look at some other observables. Do you do a better job matching the high SIF values over the corn site? If so, why, e.g.which parameter, Vcmax or LAI is mainly responsible? What temporal resolution of the parameters is necessary to capture the important variations? I suspect these questions only scratch the surface. I stress that this is potentially a good paper. What it does it does well but I believe it needs more scientific content before publication. If the authors wish to maintain it near its present form I believe it is better suited as a demonstration of a new methodology and hence to GMD.

My request is to delve a little deeper into why the system works better at some places than others. I note there seems less analysis of the Niwot Ridge results which were, in general, also less successful (lower correlation for example). Remember that a less successful assimilation is *not* a failure but rather a useful probe into model performance. It says definitively "we have a problem here and it isn't the choice of parameters". This is even clearer in this case where the parameters are allowed to vary in time.

Minor comments P14 In fact the Jacobian doesn't quite show the problem is non-linear, it could be that all the variation is a result of different forcing.

P16L3 The choice of observational error is quite important in DA, hoefuly this is checked later.

P16L10 I doubt the size of observational vector has much impact on computational efficiency, can you comment why it would?

P16L20 The choice of time resolution is also important and yours seems very short. This is likely to lead to parameters which can vary fairly rapidly in time but which are also quite uncertain as they are constrained by fewer observations. Hopefully you

can comment on whether parameters change significantly, i.e outside their uncertainty limits.

P16 Eq. 12, this should have a term from the prior included I think. Unless there's no prior.

P19L7 "reasonable and realistic" is a little vague, perhaps some references would help

P20L10 be careful about describing correlations as describing how parameters move since these are uncertainty not signal correlations. the sentence above makes it clear you understand this difference but many of your readers will be less clear.

P20L14 but here you do confuse signal and error, this correlation does NOT indicate they are changing in sync

P20 in general you seem to be quoting r^2 but claim this can be negative. You probably mean r.

P21 I'm not sure that the figures showing your algorithm works are necessary, especially in a journal like biogeosciences where you should focus more on the science and less on the algorithm.

P22L3 as noted earlier the diurnality is not a measure of nonlinearity

P22L10 don't quote improved correlation as a measure of fit, you could have a great correlation and terrible performance if, for example, diurnal variations had great phase and terrible amplitudes. rms is a better though not perfect statistic.

P25 See earlier comments on signal and error correlation.

P25 can you explain further why a strong negative correlation means you need to optimise both, the step from "you can't see them separately" to "you must do both of them" isn't so clear to me

P25 I hope you go on to compare the performance at the two sites, one of them seems

much harder than the other.

P27 I'm betting you originally tried to fit LAI at NWR and couldn't. That's not a failure, it's interesting information so is probably worth discussing. You're only fitting in 3 day windows so neither site really knows about the evolution of LAI from one window to the next so why does one work well and the other not, provided I'm guessing correctly.

P29L13 This site analysis doesn't seem as well developed as the others, e.g. quality of fit etc.

P29L30 do you mean changes in the temperature dependencies or more simply that there *is* a temperature dependence?

P30L20 In what sense is the approach "stepwise"? This term was previously used by Bacour et al. (2016, doi:xxx) to describe optimising for one observable then using its posterior parameters as priors for the next observable. They would describe your method as "all at once", what do *you* mean by stepwise?

---

## Author Comment (AC2) · 13 Oct 2018

Response to Reviewer 2 Comments

The present study attempts to develop an optimal inversion framework to use SCOPE for estimating Vcmax, m, and LAI by against measurements of carbon and energy flux from EC towers. They demonstrated the applicability of their approach in terms of capturing the seasonal variability of these key ecosystem parameters. The current work may provide additional information on estimating key ecosystem parameters from field data. Compared to the literatures, however, the novelty of the current study is not clear. There are so many papers which estimates the key ecosystem parameters

from models and EC flux tower data (e.g., Mackay et al 2012, Xu et al 2006; Wu et al 2009; Wolf et al 2006; Wang et al., 2010). What is the main novelty for this work? If it is the technical approach of an optimal inversion framework, then it may go some other technical journal. Even for the inversion framework, I didn't see too much improvements compared to previous work. My general impression is that this work is a rather technical description on the inversion framework of using SCOPE. I understood the rather detailed information by the authors, but the manuscript is really too long and some parts are too lengthy. I think some parts could be simplified.

We thank the reviewer for the comments on the manuscript. In this study we try to set up an inversion framework with the SCOPE model which is a fairly sophisticated canopy radiative transfer and energy balance model. SCOPE uses spectrally resolved irradiance and also produces spectral reflectance and fluorescence along with water and CO2 fluxes which are coupled through leaf photosynthesis and stomatal conductance. As also pointed out by Peter Rayner (reviewer 3), a major novelty of this study is breaking up the seasonal assimilation into smaller time windows and together with a fully Bayesian non-linear optimal estimation approach allows us to get posterior estimates of state vector comprising a number of important ecosystem parameters together will its full uncertainty characterization.

To address the novelty comment further we have now modified the framework to assimilate MODIS reflectance bands and match these with the spectral reflectance simulated by SCOPE model for optimal parameter estimation. We admit that our original manuscript didn't really make use of the complexity of the SCOPE model and the benefit of modeling spectrally resolved reflectance (and fluorescence, thermal emissions). A couple of retrieval examples are now presented as well for different ecosystems. The results using 2 MODIS bands indicate much better constraints on LAI and in turn on Vcmax and BBslope (which partially interfered with LAI before). This constraint further reduces the fluctuations in the retrievals due to observational noise in the fluxes. The posterior error reduction is also improved as a result. More discussion about the results

of parameter retrieval and further intercomparison within sites and year are presented as well. More discussion of physiology and ecosystem functioning as revealed with parameter estimations are included as well. Finally following the comments of all the reviewers, we have streamlined the presentation and included supporting information in the supplementary.

---

## Author Comment (AC3) · 13 Oct 2018

Response to Reviewer 1 Comments

MAIN COMMENTS

The article presents a methodology to estimate parameters of a land-surface carbon and energy balance model (SCOPE) through a Bayesian non-linear inversion framework. First, the biochemical model of photosynthesis is modified to resemble the model implemented in the land-surface model CLM4.5. Then, the values of three important parameters: Vcmax, BBslope (the slope of the empirical function relating net assimila-

tion to stomatal conductance) and LAI are computed for one growing season for three locations characterized by different vegetation types, a C4 crop, a C3 deciduous forest and a C3 evergreen forest. The methodology leads to estimate the seasonality of the analyzed parameters (Fig. 9, 14 and 18) and improve model performance in reproducing carbon and energy fluxes (Fig. 12 and 16). The article is generally well written, although with a large number of imprecisions (see detailed comments). From my evaluation the technical aspects of the inversion algorithm to estimate the parameters are rigorously implemented. However, I have a number of concerns which are detailed in the major comments below.

The authors thank the referee for the careful and detailed review of the manuscript. We consider the work presented in the manuscript as significant as it presents a framework to incorporate different types of observations at different scales and temporal resolutions in detailed process-based models to characterize explanatory dynamics of the ecosystem parameterizations which are either very uncertain or poorly represented in models.

The major objective here was to set up the inversion framework with SCOPE and demonstrate the usefulness in retrieving the temporal parameter variability. SCOPE is a detailed model about canopy physiological processes and energy balance and it can simulate the fluxes, spectral reflectance, SIF and many other variables thus offering a much greater potential. It should be noted that the parameter retrieval from a framework may not always represent actual physiology of the ecosystems. This may be due to uncertainty of the observations (e.g. GPP and LE fluxes), radiation and meteorological forcing data and others. Sometimes these datasets also represent large errors which are then propagated to the model, inversion and finally estimated parameters (sometimes even producing unphysical values). The inversion framework and results presented herein though clearly indicate that with increasing and accurate observations (constraints) we have better and more stable retrievals. We have optimized three key ecosystem parameters by allowing the parameters to take any value in the

real positive space, and in-spite of allowing such a large range we found the optimized parameters to lie within physically feasible values. This may be possible due to detailed and accurate representation of canopy processes within the SCOPE model used in the study. The novelty also lies in the windowed approach and allowing the system (parameters) to evolve over long durations. We have mainly used the GPP and LE fluxes as constraining observations but in the revised manuscript we have now included MODIS reflectance as well and included more examples which demonstrated significant enhancements in LAI and perhaps Vcmax retrievals. The incorporation of MODIS reflectances is a substantial step forward as it makes use of the uniqueness of SCOPE, i.e. the coupling of a carbon cycle model with canopy radiative transfer and full modeling of spectrally resolved reflectances as a function of ecosystem structure. Following the comments, we have substantially revised the manuscript and streamlined the presentation of some material in supplementary information. Your comments thus greatly helped improve the overall content as well as readability.

(i) The algorithm of the Bayesian inversion framework is designed to provide robust numerical results and optimize the model parameters to reduce the mismatch between observations and simulations. However, the same results are likely less meaningful from a plant physiological point of view. This is important given the premises of the authors in giving a physiological interpretation of the parameters (Page 2, LL 7-9). When I see (a) Vcmax changing from less than 10 to 80 throughout the growing season (Fig. 9 and 14), (b) LAI decreasing from 5-6 to less than 4 in the second part of the growing season for a crop, which is expected, at the very least, to maintain the same LAI until harvest (Fig. 9), or (c) a threefold variability in BBslope, I tend to think, there is a considerable adjustment of model structural issues rather than an estimation of meaningful ecosystem parameters. This is hinted by the authors when they say that they do not include evaporation from interception or have a simplified soil evaporation (Page 20, LL 2-4). Surely, Vcmax or photosynthetic capacity have been observed to change seasonally even considerably in a single tree (e.g., Wilson et al 2001; Misson et al 2006; Bauerle et al 2012; Wu et al 2017) but not in a range that cover almost the

entire variability of Vcmax globally (Kattge et al 2009). What would be the physiological explanation for such a variation in Vcmax? The seasonality of leaf nitrogen content is surely much less pronounced. I think what we see in the seasonality of parameters is largely a compensation of model structural inadequacy (e.g., the jumps in correspondence to major rainfall events) and only partially a real seasonality. Better constraints should be placed on the potential range of the parameters. Furthermore, some of these "parameter" as the seasonality of LAI could be tested against ground or remote sensing observations to support or disprove the values obtained in the optimization. It is also not surprising that model results are improved (even though it is not clear how much, since R2 results with the non-optimized model are not reported) since now the parameter space is significantly larger, being parameters allowed to vary seasonally.

The paper presents a Bayesian non-linear inversion framework with a relatively complex process-based canopy radiative transfer and energy balance model (SCOPE) to estimate three key ecosystem parameters. One of the advantages of the framework is the flexibility to incorporate different observations in the measurement vector and different variables in the state vector along with a full posterior error characterization of the state vector. It may be noted that the estimated parameters and their seasonality (in comparison to real seasonality) will include uncertainties in the process representation in the model, the uncertainty in forcing and matching observations. We also note that we retrieve the "effective Vcmax-25", while carbon cycle models typically include a stress-factor ([0-1]), which scales Vcmax to account for stress factors, which might downregulate photosynthesis. This is an important aspect, as our inversion might eventually enable us to decouple and fit the stress-factor independently under the assumption of a relatively stable "true Vcmax". In addition, we now also incorporated MODIS reflectances in our inversion scheme, which greatly reduces the variability in Vcmax and LAI and should alleviate these main concerns. We believe that parameters retrieved for the examples represent reasonable temporal/seasonal variability. Thanks to the reviewer for the insightful references we have included these in the manuscript. The reference literature indicates global variability in Vcmax is perhaps much larger

in the range of 0-150 or 200 (Gronedijk et al 2011, Archontoulis 2012), than what we have found for our study sites.

Groenendijk, M., et al. "Seasonal variation of photosynthetic model parameters and leaf area index from global Fluxnet eddy covariance data." Journal of Geophysical Research: Biogeosciences 116.G4 (2011).

Archontoulis, S. V., et al. "Leaf photosynthesis and respiration of three bioenergy crops in relation to temperature and leaf nitrogen: how conserved are biochemical model parameters among crop species?." Journal of Experimental Botany 63.2 (2011): 895-911.

Literature does support similar ranges of variability of Vcmax in the growing season (for different crop types) as we have found in our inversion results. Changes in leaf nitrogen and more importantly its allocation to Rubisco can cause substantial seasonal variability (e.g. Wilson et al 2000). There may be other biochemical reasons for increasing the photosynthetic capacity which are not parameterized in FvCB equations or even beyond scope of the current study (model structural issues). We have incorporated them in the discussion as suggested later by the reviewer in minor comments.

Section 7 (manuscript)

*"There is strong evidence from measurements that under normal conditions LAI and photosynthetic parameters have seasonal variability (Wang et al., 2008; Wilson and Baldocchi, 2000; Wilson et al., 2000) which correlate with observations of energy fluxes. Our model inversion results are in alignment and agree well with these observations. From the results, it can be seen that there is considerable seasonal variability in Vcmax and BBslope (to some extent). Previously many studies have reported measured Vcmax values of similar ranges such as, 0-70 $\mu$ mols m$-2$ s$-1$ (Wilson et al., 2000) for deciduous trees and 0- 80 $\mu$ mols m$-2$ s$-1$ as annual variability in tall Japanese red pine forests (Han et al., 2004). In addition most of the Vcmax variability is also found in systems which have seasonally variable or constant Nitrogen (N) content*

*(Wilson et al., 2000). These changes may be mostly attributed to substantial in-season changes in the fraction of total N allocated to Rubisco as well as changes in leaf mass per area (Wilson et al., 2000). In addition, most models including SCOPE have no other method of imposing environmental stress than reducing Vcmax by a stress factor [0,1]. The effect of reductions in Vcmax are a reduction in assimilation, which also suppresses transpiration (that is stress might also change BBslope). Thus, we are fitting an 'effective' Vcmax parameter, which factors in effects from true changes in Rubisco content as well as the impact of environmental stress. It may also be possible to have large seasonality variability in BBslope (15 to 25) values (Wolf et al., 2006)."*

Wilson, Kell B., Dennis D. Baldocchi, and Paul J. Hanson. "Spatial and seasonal variability of photosynthetic parameters and their relationship to leaf nitrogen in a deciduous forest." Tree Physiology 20.9 (2000): 565-578.

Throughout the description of results, we have now included both prior and posterior R2 values as well as the chi-2 error statistic which now do a much better job of comparing the results and demonstrating the improvement.

(ii) A major issue is also the use of a single year for the three locations. In this way, the robustness of such estimates across different years remain untested. In my view, an important test, would be to use the constant and seasonal parameters over few other years and see if the prediction are better in the case of seasonally variable parameters. Saying that they are better when they are optimized is completely expected and trivial. Finally, I am glad the author consider uncertainty in the flux-tower data (S$\varepsilon$), which is a very important aspect given the considerable uncertainties in flux-tower observations, however, how they do (Page 16, Line 2-3) is not very clear nor justified.

We used a single year to demonstrate the approach across three different ecosystems representing both C3 and C4 photosynthetic pathways. The three sites tested have different climates as well and we believe that the inversions across these sites test the approach and demonstrate that it is able to capture the seasonality of the parameters.

As we have shown with our examples that the predictions are better with optimized parameters over their prior constant values across different ecosystems, we are certain this will be true (to different degree) when tested for other years as well. We selected the year 2010 because it was a normal (wet) year without any significant crop stresses. In other years the vegetation may be subject to water or other stresses and there could be some deficiencies in SCOPE model to fully capture them.

Following reviewer comments, we have now incorporated MODIS reflectance bands and added a couple of more inversion retrieval examples. We have included a different year (2009) for the Ozark site as well as added another simulation for the Mead1 site for the year 2010, now including the MODIS reflectance bands. We find the reflectance acts as a much better constraint for LAI and in turn reduces fluctuations in Vcmax and BBslope to provide more realistic parameter estimates. The posterior simulations also show similar or better improvement in prediction of fluxes over their constant priors. These results have been presented in detail in the modified manuscript sections 6.3.2 and 6.4.3. The observation (flux tower) uncertainties are indeed incorporated in the Bayesian framework. In the absence of a formal error estimate from the fluxtower data itself, we currently use an average 10% uncertainty, which might be an underestimate in some cases. However, we can easily change this $S_\epsilon$ uncertainty in our retrievals and are open to suggestions from the fluxtower community. This point is further elaborated in the minor comments.

(iii) I also have doubts about the overall novelty of the study. Several other studies have been published using Bayesian approaches to parameter estimation with inverse methods in ecosystem or land-surface models (e.g., Mackay et al 2012, Xu et al 2006; Wu et al 2009; Wolf et al 2006). For instance, the authors do a remarkable job in highlighting the subtle differences of their work with respect to Wolf et al 2006 (Page 3, LL 15-21). Surely, in this article, the model is different, the optimized parameters are different and the inversion methodology has some peculiarity but overall the idea and scope is not much dissimilar from the ones of previous studies. If the novelty is

on technical aspects of the methodology, then "Biogeosciences" may not be the most appropriate venue.

The inversion framework presented here provides a generalized Bayesian non-linear inversion approach for studying the temporal dynamics of the parameterizations from observations using process-based models. We have used SCOPE here because it is fairly sophisticated canopy model which uses spectrally resolved irradiance and also produces spectral reflectance and fluorescence along with water and CO2 fluxes which are coupled through leaf photosynthesis and stomatal conductance. Along with parameter estimations the framework provides posterior error estimates of the parameters. It is important to reconcile the estimated parameters along with their posterior uncertainties measures together for physiologically explaining the temporal variability in conjunction with forcing and observations. We do agree that we haven't yet really exploited the capabilities of SCOPE in our original manuscript as the inversions presented could have been performed with a much simpler ecosystem model. Thus, we have now added MODIS reflectance data in our inversion framework, which is the first step towards making full use of the SCOPE capabilities, i.e. spectrally resolved canopy radiative transfer (including fluorescence) in the short-wave as well as thermal infrared. This addition on our revised manuscript should set our manuscript apart from previously published research.

A major novelty of the study is that the framework is flexible to adapt to temporal or spectral observations easily (which is not a feature of most available Carbon cycle models currently) and the moving window approach is well adapted for capturing the temporal variability in the parameter space. There are very limited number of studies which are aimed towards capturing the temporal dynamics of the parameters (Wolf 2006 considers temporal variability of the parameters, and differences with the current work are already indicated in the literature review). In comparison most Monte-Carlo approaches (Mackay et al 2012, Xu et al 2006; Wu et al 2009) considers time invariance of the parameters and at the max gives a range of the parameters. These

Monte Carlo methods also does not provide any definite error characterization as the Bayesian methods. This study presents a methodology by elegantly combining different ecosystem observations together to jointly estimate the temporal dynamics of ecosystem parameters using non-linear optimal approaches with process-based models. It is important to characterize the temporal variability of important ecosystem parameters in carbon-cycle or ecosystem models. The windowed approach provides an elegant approach to capturing the temporal dynamics in important ecosystem parameters and potentially more insight into stress events.

As mentioned, we have now modified the framework to assimilate observations of MODIS reflectance bands and match these with the spectral reflectance simulated by SCOPE model. A couple of retrieval examples for different ecosystems are presented. The assimilation of reflectance better constrains LAI and in turn Vcmax and BBslope. The posterior error reduction on parameters is improved as well. This is presented in detail in the revised version of the manuscript.

(iv) Finally, some of the presented material is redundant. There is an entire manuscript part in comparing new and old implementations of the biochemical model of photosynthesis (Section 2.3 Fig. 2, 3, 4 and 5), which is just relevant for the SCOPE model users, since it is quite obvious that if one substitutes temperature functions in the biochemical photosynthesis module, he/she might obtain different results. These types of analyses are carried out by any model developer all the time, without the necessity to write 5 pages of peer-reviewed paper about it. Figure 11 is also very technical and I think it would be more appropriate for an appendix or supplementary material than for a main text. The results described for the three sites are also separated and many explanatory sentences are repeated. Their presentation can be largely streamlined, combining Figure 8, 13 and 17, Figure 9, 14 and 18, and Figure 10, 15 and 19 and removing the repetitive parts, highlighting differences among the case studies, rather than iterating the overall result presentation.

We agree with the reviewer in reorganizing some parts into a supplementary material.

We would like to however keep some discussion related to the newer implementation of photosynthesis in the main text. We have modified the entire photosynthesis module together with its temperature dependence functions (which is at the heart of energy balance) as well as a new A-Ci-gs iteration implementation. It is therefore important to show that the model works (and probably shows improvement!) after the modifications. The modifications are done at the leaf element level but the comparisons are shown at the canopy level, thus the results are not entirely trivial. The main message figures 2-5 and associated text carry is that the results are consistent after the major model changes but there are many subtle and noticeable differences as well indicating room for improvement in terms of optimizing the different parameters representing the processes (lines 8-11, page 11). Following the reviewer suggestions, we have largely streamlined the manuscript, we have moved parts of the manuscript on comparing old and new photosynthesis implementations to the supplementary information along with previous figs 2, 4 and 5. We have also moved Fig. 11 to the supplementary material as suggested by the reviewer. The results section is streamlined and compactly presented with more analysis. Previous version figures 13 and 17 have been moved to the appendix as well.

DETAILED COMMENTS

Page 2 LL 5. See also Wramneby et al 2008; Pappas et al 2013.

Thank you for the references, and we have incorporated them in the manuscript.

Page 2 LL 18-19. Which model do you refer to? LSM, Ecosystem/Vegetation models or the biochemical models of photosynthesis? The first they also require shortwave radiation and precipitation as input and information about soil depth and properties, at the very least.

We refer to the ecosystem/vegetation models, which model the canopy processes to estimate the carbon and water fluxes (calculated as a byproduct of photosynthesis). SCOPE does require the specification of shortwave from which the PAR/APAR is computed. Below ground and surface processes are not simulated with SCOPE, only canopy radiative transfer and energy balance. We have modified the following lines in the manuscript to better reflect this.

*"Further, the process based canopy models require some environmental drivers such as incoming shortwave and longwave radiation, air temperature, relative humidity, wind speed, and ambient $CO_2$ concentration, along with a number of leaf and canopy parameters to simulate the fluxes of carbon in terms of gross primary production (GPP), flux of water or latent energy (LE), sensible heat (H), net radiation and others."*

Page 2. LL 26. Rate-limiting, strictly speaking, refers only to Jmax not Rd.

Thank you, we have modified the sentence to clarify this.

Page 2. LL 33-34. LAI in most of Terrestrial Biosphere Models is a prognostic variable not a parameter. As you correctly wrote in Page 3, LL 13-14. I think, this needs to be stated here.

Agreed, we have stated this here to make this point clear.

Page 3. LL 2. LAI can be also estimated from destructive observations.

Agreed and true. We have included this in the manuscript. We would like to note that in the present modeling exercise we characterize the dynamics of LAI over ecosystems and in-destructive methods seems to be the most suitable observations for comparison.

Page 3. LL 12 and Page 4 LL 12. SCOPE can model "spectrally resolved" radiation, but it remains unclear throughout the manuscript how many wavebands and which ones are considered?

A little description about wavelength ranges are already present in the SCOPE model description. The SCOPE model calculates outgoing spectrum of radiation from 0.4 to 50 $\mu$m. It resolves top of canopy incoming/outgoing radiation and reflectance in the spectral range of 0.4 to 2.5 $\mu$m @ 1nm wavebands. Further it also computes

the spectrally resolved fluorescence emission in the range of 650 to 850 nm @ 1nm wavebands. It is clarified further that SCOPE considers the fully spectrally resolved radiation in the revised manuscript as follows:

*"SCOPE resolves top of canopy incoming/outgoing shortwave radiation and reflectance in the spectral range of 400 to 2500 nm at 1nm wavelength bands. Further, it also computes the spectrally resolved fluorescence emission in the range of 650 to 850 nm at 1nm wavelength bands."*

Page 3. LL 22 See also Mackay et al 2012.

Thank you, we have included the study in the introductory paragraph.

Page 3. LL 25. I would write "often the associated computational costs. . ."

Agreed we modified the sentence as suggested.

Page 3. LL 26-31. While I completely agree that modeling SIF and comparing SIF with observations is very important, since SIF is not explicitly used in this manuscript, such a long paragraph in the introduction, deviates from the main focus of the article.

An important feature of the inversion framework we develop is the ability to bring in different data streams to investigate the dynamics of ecosystem parameters and fluxes. This is particularly relevant when we use this framework with SCOPE as it can model the fully spectrally resolved canopy reflectance as well as the fully resolved SIF emission spectrum. Although initially we have focused on carbon and water fluxes as the main input data streams for the inversion we have now revised the manuscript to include simulations with satellite (MODIS) reflectance bands to demonstrate some of this capability. We believe this is a unique advantage of using SCOPE in enabling use of data from different sensors. We feel that the description of this potential fits well in the introduction, as more SIF and spectrally resolved observations become available this framework can be utilized to better constrain and/or retrieve the dynamics of more ecosystem parameters.

Page 4. LL 26-28. So, in the end, how many prognostic temperatures do you resolve in the system? E.g., 2 temperatures for each layer for how many layers?

In SCOPE we resolve the sunlit and shaded temperatures for 60 vertical layers in the canopy. The vertical canopy is divided into 60 layers (standard implementation in SCOPE) to add up to total LAI.

Page 4. LL 28. Since you brought this up and this is not an easy problem to solve. Are iterations repeated until convergence? Which tolerance is used for convergence? If not, how many iterations are used?

Internally within SCOPE the tolerance used is the energy balance within the canopy, the iterations are repeated till convergence. The convergence criteria is an energy balance closure error <= 1.0 W/m2. If the convergence does not occur within 100 iterations, the iterations are terminated and the energy balance error is stated.

Page 5. LL 10-16. I found this part explaining differences with CLM4.5 at least awkward. As a reader I want to know what you do now, not what was different from CLM4.5 in the previous model version.

We have presented that in the existing SCOPE version the implementation of Photosynthesis and its temperature dependence are in line with the older CLM4.0. However, we update the biochemical module to make the Photosynthesis and particularly its temperature dependence fully consistent with a major Earth System Model component CLM4.5 which is improved and broadly accepted by the community. Temperature dependence was a major improvement in CLM4.5 which we have implemented in the current SCOPE version. The important and major changes made to the SCOPE model are then further listed.

Page 5. LL 20-21. Please move to an earlier part of the section the explanation of what CLM is.

We agree with the reviewer and have now included the brief description of CLM (4.5)

[Figure]

earlier in the section where CLM is mentioned for the first time.

Page 6. LL 1. There is no mention of the parameter "intrinsic quantum use efficiency" or "quantum yield of photosystem II" depending how it is expressed. While it is considered constant by most biochemical models, this could also exhibit some variability (e.g., Skillman 2008) and could have been integrated in the optimization framework.

The quantum efficiency/yield of PSII for C3 photosynthesis is mentioned in the details (Appendix equation A6). As mentioned by the reviewer the value of max PSII yield is currently assumed to be constant as 0.85 in SCOPE biochemical module and could potentially be included in the optimization framework (also like temperature dependence parameters, etc). In the present work we would like to demonstrate our approach with the three important parameters and later on work with expanded or more targeted parameter space.

Page 6. LL 4-5. As before, please delete the second part of the sentence "in the earlier SCOPE version it was implemented as potential ETR x CO2 per electron", this is not relevant here.

Thank you, we have deleted this part of the sentence.

Page 6. LL 15-20. Cs is not defined, but it must be the CO2 concentration on the leaf-surface, otherwise a leaf boundary layer resistance would be necessary in Eq. (1) and (2).

Thanks for the comment, yes Cs represents the CO2 concentration on the leaf surface we have included this in the description.

Page 6. LL 26. Iterative methods to solve the A – Ci – gs system were already included in ecohydrological models, (e.g., Ivanov et al 2008 and other land-surface models before that).

Thanks again, we have included this reference in the manuscript. In the present study we have included a Newton-Raphson scheme in SCOPE for the convergence of the A-

Ci-gs convergence. This iteration was absent (or some complex unpublished scheme was present) in previous SCOPE versions.

Page 7. LL 1. This is repetitive, it has been already stated a few times.

We have removed this sentence.

Page 7, LL 6-7 and Figure 1, Caption. It is not clear what $\pm\sigma$ variability means. It is the variability in the parameters of the temperature functions, how $\sigma$ is defined? Is the standard deviation of which parameter? What data from Leuning 2002 are considered/used for the plots? Are the temperature parameters rather than data taken from Leuning, 2002?

The parameters referred here (Fig 1) are the activation energy, deactivation energy, entropy in the temperature response functions of Vcmax (Appendix eqns B1, B2). The mean values and standard deviation ($\sigma$) are all computed from actual data of the parameters provided in the Leuning 2002 paper, the response functions are then plotted in figure 1 with these parameters. We have made the wording clearer in the caption and text to explain this part.

Page 7. Line 16. The two sites are not defined yet, their description is arriving much later in the manuscript, while it should be made upfront.

We agree that the site descriptions appear a little later in the inversion result sections, this is also clearly indicated in the parenthesis in the manuscript text. As also pointed out later the results shown in this section does not relate much to the explaining the physiological behavior of plants but rather as a comparison to show model improvement/consistency. The simulations with the SCOPE although required real observations and we show some actual comparison of GPP with tower values to show the model behavior after making changes to the model. We felt the actual meteorological/crop settings would be more relevant when looking at parameter optimization/ flux results and hence would like to present these at a later stage (result section) in the

manuscript.

Page 8. Line 5, Figure 5 and Page 11 Line 4-6. Which version has been calibrated to the data? It is not surprising that the newer implementation is closer to observations, if Vcmax or other parameters have been optimized for the newer version. For instance, in Figure 5b you could likely adjust the value of Vcmax to obtain an opposite behavior, where the old model is unbiased and the new one is.

The results presented in figures 5 and 2 only indicate the improvement/change in carbon and water fluxes due to changes/improvement made to the SCOPE model, specifically the implementations in modeling photosynthesis, its temperature dependence functions and the A-Ci-gs iterations. For these results same parameterization is used (not optimized yet but using reasonable values) and shows the difference in results obtained from old version and new modified version of the SCOPE model. What we have now moved these figures to the supplementary information, what we show is that with same parameterization but with improved process representation (photosynthesis, temperature dependence, etc) the model does a better job in capturing the fluxes.

Page 8. Line 9-10 and Figure 3 and 4 caption. I do not think you are showing any result for the Missouri Ozark site or Nebraska-Mead-1, you just use the meteorological forcing and C3 vegetation type derived for these sites to run the two version of the biochemical model and make a comparison. Strictly speaking you could have done a test varying temperature, VPD, and PAR for C3 and C4 vegetation without referring to any specific site. However, I do not find this part generally insightful for a journal article.

We partially agree, figures 3 and 4 (previous version) just illustrate the difference between two (old and new) model implementations, but we would like to argue that these results are a part of the model comparisons which also includes figures 2 and 5 (previous) where we show site specific data such as GPP and LE comparisons. We therefore believe that it may be relevant to show the comparisons with real site-specific meteorological forcing data as well. Also, it may be erroneous to generate all the meteorological

forcing synthetically as it may generate combinations of forcing variables (such as T, VPD or RH) which are unphysical and this may also lead to unphysical model outputs when driven with these forcing. Finally, we believe that the comparisons are necessary and important, we have modified the entire photosynthesis module together with its temperature dependence functions (which is the heart of energy balance) as well as a new A-Ci-gs iteration implementation. It is therefore important to show the model works (and probably shows improvement!) after the modifications. The modifications are done at leaf element level but the comparisons are shown at the canopy level. The messages figures 2-5 (previous) conveyed are that the results are consistent after the major model changes but there are many subtle and noticeable differences as well indicating room for improvement in terms of optimizing the different parameters representing the processes (lines 16-19, page 9 revised version). This description has now been streamlined and figures 2, 4 and 5 have been moved to the supplementary information.

Page 13. Line 4. The Jacobian depends on the linearization point, Xl, which is somehow approximated for any optimization window. This is discussed later but maybe it should be stated here already.

We have already briefly stated that the Jacobian for SCOPE is estimated numerically here, the formulation of moving window are stated later and therefore the method of setting up and estimating the Jacobians are also stated along with it.

Page 13. Line 14-15. It is fine to have a more in-depth treatment in the appendix, but at least you need to define the terms, Ki (the Jacobian matrix) is never defined at this stage of the article.

Thank you, we have defined Ki here at this step.

Page 14, Figure 6, and also Page 22, LL 5-6. Some of the derivative values are a bit surprising. Why LE is decreasing with increasing LAI? Is due to self-shading effects? This is generally counterintuitive. Even more difficult to understand is why

LE decreases with increasing Vcmax. Higher photosynthesis should lead to higher stomatal conductance and thus higher LE, especially in a model like SCOPE. These unexpected behaviors need justification.

The decrease in LE with an increase in LAI may indeed be due to self-shading effects, but we can't say for sure. We agree that some of these derivatives appear counter-intuitive from a leaf-level perspective but we obtain these from the SCOPE model and obtain stable inversions. One reason might be the full coupling of the energy balance, as leaf (and canopy) temperature changes when parameters like Vcmax are changed. In general, the nature of the Jacobians also depends on the meteorological conditions (temperature, VPD, RH etc) and their integration from leaf element level to canopy level is non-linear as such we expect the Jacobians to be diverse and maybe counterintuitive. As stated in the text: "this diversity and nature of the Jacobians allows us to explore the optimality of the many parameters together representing linked non-linear physiological processes within the canopy".

Page 15. Figure 7 caption. You need to specify what is "m" and what is intended here for $\Delta X$. Plus, it should be better stated what the measurement vector refers to, since the derivative must be computed with model outputs.

Thank you, we have modified the figure caption to reflect and better explain the above-mentioned points.

Page 16. Line 2-3. The observational error matrix $S_\epsilon$ is quite important given the general uncertainty of flux-tower observations. However, from such a brief description "using noise standard deviation as 10% of observations" is not clear how this is computed. Does it mean that you assume an uncertainty of 10%? This is likely quite a low number in the context of flux-tower observations (e.g., Leuning et al 2012).

Thank you, our Bayesian inversion framework includes both observation noise and parameter prior errors which can be adjusted as per different site conditions to better optimize the retrievals. We assume that the flux observations have uncertainty of 10%.

The actual uncertainty of flux observations is hard to characterize and also not available readily that is why we have made this assumption. The surface energy balance closure error has been generally reported to be around 10-30% (Wilson et. al, 2002, Von Randow et. al, 2004, Sanchez et. al 2010) and is found to be dependent on time-scales due to differences in energy storage terms in ecosystems (Reed, et. al 2018) (Leuning et. al. 2012 specifies closure for 8% of sites sites with half hourly datasets). However, the important point we demonstrate here is the feasibility of the approach in parameter retrieval with full posterior error characterization using suitable apriori uncertainties. We have included some discussion to state this point and the scope of characterizing the observation noise better in the framework.

Reed, David E., et al. "Time dependency of eddy covariance site energy balance." Agricultural and Forest Meteorology 249 (2018): 467-478.

Wilson, Kell, et al. "Energy balance closure at FLUXNET sites." Agricultural and Forest Meteorology 113.1 (2002): 223-243.

Sánchez, J. M., V. Caselles, and E. M. Rubio. "Analysis of the energy balance closure over a FLUXNET boreal forest in Finland." Hydrology and Earth System Sciences 14.8 (2010): 1487-1497.

von Randow, Coauthors, et al. "Comparative measurements and seasonal variations in energy and carbon exchange over forest and pasture in South West Amazonia." Theoretical and Applied Climatology 78.1-3 (2004): 5-26.

Page 16. Line 5. Which iteration step? The current one? This is relevant since K depends on where it is computed.

Yes, the Jacobian matrix K is computed at every iteration step, we modified the text to make this clear.

Page 16. Line 6-7. I am not sure I can see very well the concatenation between observations Y and modeled values F(X) in Figure 7. I just see $\triangle$Y.

The concatenation is performed for different variables (GPP, LE, etc) represented in different colors in the figure, it is done for both observations (tower fluxes) and modeled values separately, $\Delta Y$ represents the difference in observations and modeled values. We have clarified this in the figure caption and also the associated text.

Page 16. Line 20-22. I agree with the authors that 3-days sounds as a reasonable length for separating the time scales of parameter variability and meteorological forcing. But, what does it happen if you modify the time window and instead of 3 days you select 7 days or 15 days? Do you get similar seasonality of parameters? This is a test, which could be relevant in the scope of this article.

For the present problem 3-day sounds a reasonable length of the time-window, some early and initial testing with daily, 2-day windows also captured the seasonality. Windows of 7 or 15 days may be a little too long especially if there are artifacts in the forcing or observations data (as we have seen in many cases) it will make our parameter estimations poorer. The results will also depend on the parameter set we are optimizing, for the current set (Vcmax, BBslope, LAI) 3 days seems an appropriate window. We have not tried longer time-windows in our setup yet but assume they will work equally well if the boundary conditions (T/VPD, etc) don't change dramatically over the time-window. This might be something to test in the future.

Page 16. Line 24. I do not find where this is mentioned in Section 5.

This point is briefly mentioned (Details are in Rodgers 2000 which is referred in the article). The detailed criterion is: The strategy to update $\gamma$ is that we compute R which is the ratio of the true change in cost function to the expected change in cost function. If R > 0.75 reduce $\gamma$ (factor of 2), update X. If R < 0.25 increase $\gamma$ (factor of 10), do not update X.

Page 17. Equation (17). The subscript "jj" is undefined.

Since S or Sa is a matrix, the index "jj" refers to the diagonal elements here, have

clarified this point in the text.

Page 18. Figure 8. Did you see that incoming shortwave radiation is reconstructed and actually almost equal for the first 60 days? Afterwards, fortunately, you do not use this period because it is not in the growing season, but such type of artifacts, which are frequent in Fluxnet data, could jeopardize your procedure. This may need a mention.

Thank you, artifacts in the flux data influence the inversion and optimization greatly and affects the results. We added the following lines of discussion on this point. *"Apart from observations, it should also be noted that filtering and quality control may be necessary for the meteorological and/or forcing fluxes as artifacts in the data can influence the inversion and optimization and greatly affects the results."*

Page 18. Line 8. Why did you select a single year? I think it would be rather important to see how seasonality of parameters is retrieved in different years.

We used a single year to demonstrate the approach across three different ecosystems representing both C3 and C4 photosynthetic pathways. The three sites tested have different climates as well and we believe that the simulations across sites does test the approach and demonstrate that it is able to capture the seasonality of the parameters. It is also almost certain that seasonally variable parameters over constant values will show improvement in prediction over any other years as well. We selected the year 2010 because it was a normal (wet) year without any significant crop stresses. With crop stresses model structural deficiencies (also in SCOPE (e.g in the stomatal conductance model or absence of plant hydraulics representation) may appear thus hindering the retrievals. Through the results across different ecosystems we demonstrate the approach works in terms of capturing seasonality of important ecosystem parameters which otherwise are often considered as constant values in the models. Following the reviewer suggestions, we have also added further support to our findings by including the simulation for the year 2009 for the Ozark site as well as including the MODIS reflectance in the inversion. The parameter retrievals are found to be better

constrained but having similar variability across the season. The results are presented in detail in revision.

Page 19. Line 8-9, Please provide references for such changes in BBslope, LAI and Vcmax if they are found to be reasonable, which I do not think it is the case for BBslope and Vcmax.

We believe the temporal variability in Vcmax (and order of magnitude) presented in the manuscript is very much feasible. This is already stated above in response previously and we have cited the proper references as well following the suggestions.

Wilson et al 2000 reported values of 0-70 $\mu$mols m-2 s-1 for Vcmax (Fig 2) for the deciduous trees also similar large ranges are reported in figs 3-5.

Han et al 2004 reported measurements of Vcmax over an entire year in the range of 0-80 $\mu$mols m-2 s-1 (Fig. 3)

Gronedijk et al 2011 presented the seasonality of Vcmax over cold, warm and temperature regions for grassland, savanna, broadleaf deciduous and evergreen forests. The ranges over a year vary from 0-150 $\mu$mols m-2 s-1 (fig, 5).

Archontoulis 2012 also presented Vcmax experimental values for crops ranging from 25 to 150 $\mu$mols m-2 s-1.

It may also be possible to have large variability in BBslope values Wolf et al 2006 showed that seasonal variability in BBslope could vary from 15 to 25 (fig 7a).

Page 20. Line 12 and captions of Figure 10 and also Figure 15 and Figure 19. If it is a "correlation coefficient", why the symbol r2 is used, this is typically reserved for the "coefficient of determination", and not for the correlation coefficient, which is typically indicated by r.

The correlation mentioned here is the correlation in terms of two random variables (x,y), this is defined as COV(x,y)/( $\sigma$x $\sigma$y). We have changed it to more standard symbol such

as $\rho$ throughout the manuscript.

Page 21. Figure 11. The x-label with the indexes is not defined in the figure caption but only in the main text.

The x-label is an index representing the sequence (of values) after the observations are concatenated. We have indicated the meaning of the indices in the caption.

Page 22. LL 7 and LL 10 and Figure 12 caption. How much is it this "net improvement"? The value of R2 for the previous parameter set is not reported, values are only reported for the optimized model.

We have computed the value of R2 for the unoptimized parameter set and added them for every example in the paper which makes the improvement clear. In addition, we have also reported the chi-square statistic for comparing the results.

Page 23. LL 12. It is quite well-known that the growing season is longer for a deciduous forest than for a crop. I would suggest eliminating "we find".

Thanks, we have made the correction.

Page 24, LL 13 and Page 29, LL 6-7. What could justify a threefold change of BBslope in a single growing season? This is theoretically an intrinsic property of the stomatal regulation, I can see how can change during leaf development (very first part of the growing season) or if water stress occur, but I do not see what can justify such a large variability throughout the entire season.

The BBslope results presented (just with flux observations) does not show three-fold change everywhere, for the deciduous site at Ozark the BBslope changes between DOY 100 and 250 are very subtle and small with the values varying between 5-7. There is a sudden rise after DOY 250 but the error reduction (uncertainty) measure also indicates there could be some error in the retrieval after DOY 250. For the Evergreen forests the three-fold variability could be representative of the stomatal activity after a dominant cold season and it could be a model structural issue as well (as now

Interactive
comment

described therein). For the mead crop again the general variability is between 5-15, at the peak growing season some of the reasons for overestimation around DOY 220 due to evaporation from wet canopies has already been presented (lines 1-5 page 20 in the text). Moreover Fig 8 also represents that the values around these times are more uncertain compared to other times of the year. Finally, it may be possible for large variability in BBslope values Wolf et al 2006 (Fig 7a) showed that seasonal variability in BBslope could vary from 15 to 25. The newer simulations and comparisons with MODIS data also reflect that it could be much better constrained in our case the improvement is clear for the mid-season spike for the Mead1 site

Page 25. LL 13-16. The value of R2 for the previous parameter set is not reported, values are only reported for the optimized model. How much the results were improved remains unverifiable, although clear from Figure 16.

Thank you, yes, we have added the value of R2 for the unoptimized case as well to make the improvement clearer.

Page 27. LL 2-6. Please consider that at Niwot Ridge snow-cover is affecting energy exchange and potentially GPP for a large fraction of the year. It is true that you focus only in the snow free-period, but this confounding element needs to be stated somewhere.

Thank you, this point stated in the manuscript where the Niwot ridge site is discussed.

Page 27. LL 10. The constraint on LAI from observation is a very good addition to the modeling exercise, I would have liked to see this type of constraints placed also for other sites, or parameters, whenever available.

Thank you, the constraint on LAI for Niwot ridge was available from literature, in comparison constraints on Vcmax or BBslope are much harder to obtain. We have now added MODIS data as well to enable better constraints at the other sites.

Page 29. LL 16 and Page 30 LL 4-5. I am sorry, but I am thinking you mostly re-

trieve "model parameters" and not "ecosystem parameters", and I also tend to think you are "overfitting" the SCOPE model rather than constraining it. The advantage of using seasonality of parameters for predictive simulations (e.g., in other years or other conditions) remain to be tested.

We would like to argue that the retrieved model parameters which are representative of ecosystem parameters. We have utilized the fluxes at ecosystem level from flux towers in process-based canopy models (assuming homogeneity in ecosystems). The results obtained from the inversion are thus representative of the ecosystem parameters with certain uncertainty. Some information about overfitting may be examined from the chi2 values computed during the inversion (which are not presented in the manuscript) but the reduction in chi2 values (posterior simulation) after optimization indicate that these points may not be overfitted. The posterior error reduction of the retrieved parameters also indicates a clear error reduction in all parameters. Some testing about predictive ability in other years is also provided with simulation at Ozark with reflectance constraint for the year 2009. The method is also further tested for different climatic conditions and vegetation types across different ecosystems.

Page 29. LL 22. Yes, LAI and Vcmax have seasonal variability but for Vcmax unlikely in the order of magnitude that is presented here.

The optimized parameters also include the uncertainty associated with observations as well as errors in the process representations of other unoptimized parameterizations in the model. The variability of Vcmax we have obtained is not highly unlikely (as mentioned in the previous comments and supported with references), our optimization and inversion framework also account for full posterior error characterization and test of independence (correlation coefficients). The results of optimized Vcmax must also be therefore interpreted with these uncertainty measures. As presented earlier the variability in Vcmax may be due to seasonal variability in relocating nitrogen to Rubisco and further other stress factors which may be lumped into the Vcmax values. Please note that the addition of MODIS reflectances has now greatly reduced the variability in

LAI and Vcmax.

Page 29. LL 27-30. While interesting the discussion on SIF is out of place, since SIF is not used or treated in this article.

We have already extended our work by incorporating satellite reflectance in the inversions. We would like to mention SIF here because it is an important output from SCOPE model (spectrally resolved) and the inversion framework is easily adaptable to incorporate these observations which is an important future research direction.

Page 30. LL 16. Optimality approaches (e.g., Medlyn et al 2011, Katul et al 2010) do not typically comprise soil-moisture dependences, which therefore need to be included as additional parameterizations.

We agree and indicated this point in the discussion.

Page 30. LL 18. Rather than "better solution", I would say it can provide "optimized model parameters."

We have changed the wording as suggested.

Page 30. LL 19-25. All this paragraph is emphasizing the technical aspects of the "optimal inverse estimation of the parameters", I am wondering if this is the most effective way of concluding the manuscript.

We have revised this to include physiological aspects into the discussion.

Page 31. LL 7-8. This sentence is quite repetitive.

We have removed the sentence.

Page 34. LL 9. I would suggest to write "will allow us".

Thank you we have made the change.

---

## Author Comment (AC4) · 13 Oct 2018

Response to Reviewer 3, Peter Rayner's Comments

This paper demonstrates a method for assimilating site-level flux observations into a terrestrial biosphere model. Its novelty lies in breaking the assimilation into short windows to capture high-frequency variations in the parameters it estimates. given the variety of journals within the Copernicus family, I wonder whether this article is better suited to GMD than BG (see comments below) but this is mainly a question for the editor. The paper is also clearly written, verging on the tutorial at times.

[Figure]

I have one significant concern with the paper and one general request for more analysis. My concern is the analysis of the results. This is quite thin. The only commentary I can see on the results in the discussion section is: "There is strong evidence from measurements that under normal conditions LAI and photosynthetic parameters have seasonal variability [Wang et al., 2008; Wilson and Baldocchi, 2000; Wilson et al., 2000] which correlate with observations of energy fluxes. Our model inversion results are in alignment and agree well with these observations." this seems quite a poor scientific return from a difficult and well-executed piece of work. I would recommend particularly using the posterior simulation to look at some other observables. Do you do a better job matching the high SIF values over the corn site? If so, why, e.g. which parameter, Vcmax or LAI is mainly responsible? What temporal resolution of the parameters is necessary to capture the important variations? I suspect these questions only scratch the surface. I stress that this is potentially a good paper. What it does it does well but I believe it needs more scientific content before publication. If the authors wish to maintain it near its present form I believe it is better suited as a demonstration of a new methodology and hence to GMD.

My request is to delve a little deeper into why the system works better at some places than others. I note there seems less analysis of the Niwot Ridge results which were, in general, also less successful (lower correlation for example). Remember that a less successful assimilation is *not* a failure but rather a useful probe into model performance. It says definitively "we have a problem here and it isn't the choice of parameters". This is even clearer in this case where the parameters are allowed to vary in time.

We thank Peter Rayner for his valuable and insightful comments. These comments have helped us to improve our manuscript greatly. We have addressed the concerns raised and made major revisions to the manuscript following the comments of all the reviewers. We would like to publish this work in Biogeosciences as we think that BG will be a great outlet for readers interested in the coupled carbon and water dynamics

in ecosystems. The developed moving window inversion framework would serve as a valuable tool for the exploration of different and rather difficult to measure ecosystem parameters using a number of observational data streams, this study is an initial step towards this. Moreover, the SCOPE model, which is in the core of inversion framework is also published in Biogeosciences. We thus hope that this journal is appropriate, especially given our substantial improvements in the revised version.

We have added significantly more analysis to the manuscript. We have now incorporated MODIS spectral reflectance bands in the inversion framework for two of our examples. The results are promising and suggest much better constraint on LAI, which in turn reduced fluctuations in Vcmax and BBslope. The retrieved parameters are more realistic and the sensitivity of the inversion towards sudden fluctuations in tower observations is reduced. Within and inter site comparison of the retrievals is also presented when reflectance data is assimilated in the inversion framework.

We have also further added more posterior simulation results and discussed the effect of optimization results for the different sites for different years with and without using MODIS data with flux observations. As suggested we have gone deeper to better explain retrieval results and their fluctuations from the simulations. Discussion about effects of nitrogen variability and Rubisco allocation on Vcmax seasonal variability is also presented. We have also analyzed the results of posterior simulations from Niwot ridge as suggested. Further, we have streamlined our work by moving some material to the supplementary information.

Minor comments P14 In fact the Jacobian doesn't quite show the problem is non-linear, it could be that all the variation is a result of different forcing.

We would like to clarify that the Jacobian shows the response slope at different times of the day due to a small perturbation in the variables of the state vector. Most importantly, we find the Jacobian to changes with subsequent iterations, which is a clear sign of non-linearity.

P16L3 The choice of observational error is quite important in DA, hopefully this is checked later.

Yes, we have included the observational error in the inversion scheme. We assume that the flux observations have uncertainty of 10%. The actual uncertainty of flux observations is hard to characterize and also not available readily, which is why we have made this simplified assumption. The surface energy balance closure error has been generally reported to be around 10-30% (Wilson et. al, 2002, Von Randow et. al, 2004, Sanchez et. al 2010) and is found to be dependent on time-scales due to differences in energy storage terms in ecosystems (Reed, et. al 2018). However, the important point we demonstrate here is the feasibility of the approach in parameter retrieval with full posterior error characterization using suitable a-priori uncertainties. We have included some discussion to state this point and the scope of characterizing the observation noise better in the framework.

Reed, David E., et al. "Time dependency of eddy covariance site energy balance." Agricultural and Forest Meteorology 249 (2018): 467-478.

Wilson, Kell, et al. "Energy balance closure at FLUXNET sites." Agricultural and Forest Meteorology 113.1 (2002): 223-243.

Sánchez, J. M., V. Caselles, and E. M. Rubio. "Analysis of the energy balance closure over a FLUXNET boreal forest in Finland." Hydrology and Earth System Sciences 14.8 (2010): 1487-1497.

von Randow, Coauthors, et al. "Comparative measurements and seasonal variations in energy and carbon exchange over forest and pasture in South West Amazonia." Theoretical and Applied Climatology 78.1-3 (2004): 5-26.

P16L10 I doubt the size of observational vector has much impact on computational efficiency, can you comment why it would?

Thanks for the comment, by size of the observational vector we meant the number of

days and the number of time points (half hourly/hourly) for constructing the concatenated observation vector. This will have a direct impact because it will increase the number of time points (instances) for the forward model runs this will also increase with the number of parameters. This will significantly increase the computational time.

P16L20 The choice of time resolution is also important and yours seems very short. This is likely to lead to parameters which can vary fairly rapidly in time but which are also quite uncertain as they are constrained by fewer observations. Hopefully you can comment on whether parameters change significantly, i.e outside their uncertainty limits.

Thanks for the comments, we believe that the 3-day window sounds is short but reasonable, as variations in environmental stress can happen on synaptic time-scales. It might eventually be better to group the season into blocks with similar "drivers", e.g. VPD, temperature, PAR, but for now, we tried to find a consistent window length, which would allow us to discern short-term fluctuations. We had performed some initial testing with other window sizes (not shown), which showed that a 3-day window is appropriate. Regarding the rapid and abrupt changing of parameters, we could observe this in particular when flux observations appeared to be quite noisy. The addition of more constraint in these cases is extremely beneficial as we have now clearly demonstrated with the inclusion of MODIS reflectance data, which has greatly improved our results for the Mead-1 site. The measure of error reduction has also significantly improved with the inclusion of reflectance data.

P16 Eq. 12, this should have a term from the prior included I think. Unless there's no prior.

Thank you, the full chi-square error we are minimizing in the optimization does have a term from prior error included. However, in order to test the convergence for each iteration in the retrieval windows (and stopping criteria) we use the criterion given in Eq. 12 to test the difference between the fit and measurements (excluding the prior).

This is clarified in the manuscript.

P19L7 "reasonable and realistic" is a little vague, perhaps some references would help.

We agree and have removed this sentence and we have included references in the discussion of results for our examples which support the values of Vcmax and other parameters obtained in this study.

P20L10 be careful about describing correlations as describing how parameters move since these are uncertainty not signal correlations. the sentence above makes it clear you understand this difference but many of your readers will be less clear.

Thank you for this comment/warning. Yes we agree that the error correlation has to be considered as a posterior inversion property and not be confused with actual physical behavior of the variables. These are a result of the inverse retrievals and gives us an indication whether the variable pairs are independent or have a positive or negative association in the retrievals. The actual association in nature between the variables may or may not be similar. We have further clarified this in the manuscript.

P20L14 but here you do confuse signal and error, this correlation does NOT indicate they are changing in sync

We agree the relationship between the variables is only valid in the context of the retrievals and this may or may not be the true association between the variables we find in nature and this association may change depending on different environmental stresses and conditions. We will probably have more confidence regarding the true nature of correlation between the variables if under different retrieval schemes/constraints the error correlations are found to be similar. We are able to include some discussion when we incorporate the MODIS reflectance in the error covariances and compare the error correlations between the two cases. MODIS reflectance data greatly helps to reduce error correlation, in particular between Vcmax and LAI.

P20 in general you seem to be quoting r2 but claim this can be negative. You probably

mean r.

This was also mentioned by the other reviewer, we have now changed all instances in the text and figure to $\rho$ for the correlation coefficient.

P21 I'm not sure that the figures showing your algorithm works are necessary, especially in a journal like biogeosciences where you should focus more on the science and less on the algorithm.

Thank you, we as also mentioned by the other reviewer we have streamlined the presentation and moved some parts to the supplementary information and appendix. We have now moved figure 11 to the appendix as well.

P22L3 as noted earlier the diurnality is not a measure of nonlinearity.

Thank you, as we mentioned we think the variability throughout the day and subsequent evolution when multiple days are concatenated together makes the problem non-linear. In most cases the variability seems diurnal but in some cases K matrix becomes a highly variable and represents a non-linear 3-d surface which varies with different environmental conditions.

P22L10 don't quote improved correlation as a measure of fit, you could have a great correlation and terrible performance if, for example, diurnal variations had great phase and terrible amplitudes. rms is a better though not perfect statistic.

Thank you, we have now presented the coefficient of determination for both prior and posterior simulations for all the examples, together with this we have also now presented the chi2 error statistic as a measure to represent the improvement in performance.

P25 See earlier comments on signal and error correlation.

Thank you, we have again clarified this part in the manuscript.

P25 can you explain further why a strong negative correlation means you need to

optimise both, the step from "you can't see them separately" to "you must do both of them" isn't so clear to me.

Thank you, in this context we simply mean that from the retrieval framework perspective the results indicate there is a strong negative linear association between the two variables. As such these are not independent and therefore not ideal to be optimized independently. We agree that the logic as written was faulty and we have removed this from the text.

P25 I hope you go on to compare the performance at the two sites, one of them seems much harder than the other.

We have now included the MODIS datasets and provided a substantial comparison of the results of parameter retrievals using only flux observations to that using both flux and reflectance observations. The variability in all the three parameters greatly reduced due to the addition of reflectance for the Mead site and at the Ozark site this change made the parameters to be more realistic in addition the posterior simulations suggested a significant improvement in LE fluxes over the other year. These are presented in detail in the revised manuscript.

P27 I'm betting you originally tried to fit LAI at NWR and couldn't. That's not a failure, it's interesting information so is probably worth discussing. You're only fitting in 3 day windows so neither site really knows about the evolution of LAI from one window to the next so why does one work well and the other not, provided I'm guessing correctly.

Thank you, yes this is the unique advantage of using a fully Bayesian framework, we found out approximately the true expected value of LAI for NR and prescribed an extremely low prior error on it and this our windowed simulations maintain it as nearly constant values, this is also a nice test about the mechanism of the inversion. We did some tests where the LAI was allowed to vary and it did indeed trying to match mainly the GPP and LE variability. In terms of physiology the changes in GPP (and LE) in SCOPE is mostly attributed to LAI and Vcmax (and BBslope). In Niwot ridge due to

cold climates there is other plant physiological signaling which stops the photosynthesis without apparent changes in LAI (or Vcmax) like the deciduous forests. This is thus probably a model structural issue which SCOPE is not able to capture just as a stress factor in Vcmax. We have added this part in the NR discussion of results.

P29L13 This site analysis doesn't seem as well developed as the others, e.g. quality of fit etc.

We have now included the results of posterior and prior simulations and discussed the fits after optimizing the parameters.

P29L30 do you mean changes in the temperature dependencies or more simply that there *is* a temperature dependence?

Thank you, we mean changes in temperature dependency due to changes in activation, deactivation and entropy parameters which are incorporated carefully into the modified version of SCOPE in this study and which the current inversion framework is fully equipped to optimize. As discussed this is although a future scope of work.

P30L20 In what sense is the approach "stepwise"? This term was previously used by Bacour et al. (2015), doi:10.1002/2015JG002966) to describe optimising for one observable then using its posterior parameters as priors for the next observable. They would describe your method as "all at once", what do *you* mean by stepwise?

We meant stepwise in the context of a within window optimization, in a sense that the LM algorithm takes a stepwise change in the parameter space taking into account the prior and the observation errors to achieve optimal solutions. As pointed when we look at the broader scheme of things seasonally the optimization seems to be all at once for each time window. We have slightly modified the sentence to better present this.

---

## Author Comment (AC5) · 17 Oct 2018

General Response to all Reviewer Comments

Many thanks to the reviewers for a careful, detailed and valuable review of the manuscript. The comments have helped us to greatly improve the quality of our manuscript. The overall major issues raised by the reviewers were regarding (i) the overall novelty of the study, (ii) more analysis and corresponding discussion on variability of retrieved parameters and (iii) streamlining the presentation.

We have addressed all of these issues in detail and made significant revisions to our

manuscript.

(i) The major novelty of this study lies in developing a flexible windowed fully Bayesian framework for assimilating a number of different constraining data-streams (such as carbon and energy fluxes, spectral reflectance, etc) for estimating the seasonal variability of a number of important ecosystem parameters and improve model prediction performance. In order to take a step forward and demonstrate the complexity and uniqueness of using SCOPE model (in modeling fully resolved spectral reflectance and showing the linkage between coupled carbon/water cycle and canopy radiative transfer) within the framework and clearly distinguishing our work from previous research, we have now included 2 MODIS reflectance bands in our inversion framework. The results indicate that reflectance constrains LAI better which in turn reduced the fluctuations in Vcmax and BBslope leading to more realistic parameter estimates. We have demonstrated the improvement in results with simulations of our study sites.

(ii) More analysis and discussion are presented as to the inclusion of MODIS reflectance along with flux data in the revised manuscript. We have also highlighted the possible connection to seasonal variability in leaf nitrogen and fractional allocation to Rubisco to explain the seasonal variability of parameters from the retrievals (Vcmax, BBslope) and their ranges from available literature.

(iii) We have simplified the model inter-comparison of previous and newer implementation of Photosynthesis. The result and discussion section are also streamlined as suggested. We have moved some details and figures to the supplementary information.

The detailed response to each individual reviewer comment is presented separately in the response to reviewers.